# Mitotic replisome disassembly depends on TRAIP ubiquitin ligase activity

Sara Priego Moreno*, Rebecca M Jones*, Divyasree Poovathumkadavil, Shaun Scaramuzza, Agnieszka Gambus

**We have shown previously that the process of replication machinery (replisome) disassembly at the termination of DNA replication forks in the S-phase is driven through polyubiquitylation of one of the replicative helicase subunits (Mcm7) by Cul2[LRR1] ubiquitin ligase. Interestingly, upon inhibition of this pathway in *Caenorhabditis elegans* embryos, the replisomes retained on chromatin were unloaded in the subsequent mitosis. Here, we show that this mitotic replisome disassembly pathway exists in *Xenopus laevis* egg extract and we determine the first elements of its regulation. The mitotic disassembly pathway depends on the formation of K6- and K63-linked ubiquitin chains on Mcm7 by TRAIP ubiquitin ligase and the activity of p97/VCP protein segregase. Unlike in lower eukaryotes, however, it does not require SUMO modifications. Importantly, we also show that this process can remove all replisomes from mitotic chromatin, including stalled ones, which indicates a wide application for this pathway over being just a "backup" for terminated replisomes. Finally, we characterise the composition of the replisome retained on chromatin until mitosis.**

## Introduction

Faithful cell division is the basis for the propagation of life and requires accurate duplication of all genetic information. DNA replication must be precisely regulated as unrepaired mistakes can change cell behaviour with potentially severe consequences, such as genetic disease, cancer, and premature ageing (Burrell et al, 2013). Fundamental studies have led to a step change in our understanding of the initiation of DNA replication and DNA synthesis, but until discovery of the first elements of the eukaryotic replisome disassembly mechanism in 2014 (Maric et al, 2014; Moreno et al, 2014), the termination stage of eukaryotic replication was mostly unexplored.

DNA replication initiates from thousands of replication origins. They are the positions within the genome where replicative helicases become activated and start unwinding DNA while moving in opposite directions, away from each other, creating two DNA replication forks.

The replicative helicase is composed of Cdc45, Mcm2-7 hexamer, and GINS complex (CMG complex) (Moyer et al, 2006); it is positioned at the tip of replication forks and forms a platform for replisome assembly (Replisome Progression Complex) (Gambus et al, 2006). Once established, the replication forks replicate chromatin until they encounter forks coming in opposite directions from neighbouring origins. At this point, termination of replication forks takes place. As CMG helicases travel on the leading strand templates at the forks, the strand encircled by converging helicases differs because of the antiparallel nature of the DNA molecule (Fu et al, 2011). The two converging helicases can therefore pass each other, allowing for completion of DNA synthesis. Finally, removal of the replisome from fully duplicated DNA is the last stage of termination of forks (Dewar et al, 2015). We have shown that in *Xenopus laevis* egg extract and in *Caenorhabditis elegans* embryos, this replisome removal in S-phase is driven by Cul2[LRR1] ubiquitin ligase, which ubiquitylates Mcm7 within the terminated CMG complex (Sonneville et al, 2017). Such modified CMG is then recognised by p97/VCP segregase and removed from chromatin allowing for disassembly of the whole replisome built around the helicase (Moreno et al, 2014).

Most notably, we have shown that in *C. elegans* embryos, any helicase complexes that fail to be unloaded in the S-phase are alternatively unloaded in the prophase of mitosis (Sonneville et al, 2017). This potential backup mechanism can be detected when CUL-2[LRR-1] activity is blocked and, like S-phase pathway, depends on the p97 segregase for unloading. Unlike the S-phase pathway, however, it requires an additional p97 cofactor UBXN-3/FAF1 and the SUMO-protease ULP-4 (Senp6/7 homologue in higher eukaryotes) (Sonneville et al, 2017). Interestingly, budding yeast do not possess this mitotic replisome disassembly pathway; cells lacking SCF[Dia2] activity, the ubiquitin ligase responsible for Mcm7 ubiquitylation in *Saccharomyces cerevisiae*, accumulate post-termination replisomes on DNA until the next G1 of the next cell cycle (Maric et al, 2014). Our aim, therefore, was to determine if this mitotic replisome disassembly pathway is functioning in higher eukaryotes or if it is a phenomenon specific to *C. elegans* embryos. Here, we show that a mitotic replisome disassembly pathway does exist in *X. laevis* egg extract and determine the first elements of its

Institute for Cancer and Genomic Sciences, College of Medical and Dental Sciences, University of Birmingham, Birmingham, UK

Correspondence: a.gambus@bham.ac.uk
Sara Priego Moreno's present address is Salk Institute for Biological Studies, La Jolla, CA, USA
*Sara Priego Moreno and Rebecca M Jones contributed equally to this work

regulation. We show that only a restricted part of the replisome stays retained on chromatin through into mitosis in *Xenopus* egg extract. The disassembly of this replisome is independent of Cullin-type ubiquitin ligases but requires p97 segregase function. Mitotic replisome disassembly depends on K6- and K63-linked ubiquitin chains but not SUMO modifications. In addition, we show that stalled forms of helicase can also be unloaded using the same mechanism, suggesting that rather than being a backup pathway for the disassembly of terminated replisomes, this process is essential to remove any replisome from chromatin before cell division. Finally, we identify TRAIP ubiquitin ligase as essential for Mcm7 ubiquitylation and replisome disassembly in mitosis.

## Results

*X. laevis* egg extract is a cell-free system, which has proven to be instrumental over the years in studies of DNA replication. *Xenopus* egg extract contains stockpiles of cell cycle factors which support efficient replication of DNA templates in vitro, with the recapitulation of most of the biochemical reactions that take place in living cells. To retain high synchronicity in our system, we restrict the replication reaction in the extract to only one round through blocking protein synthesis with cycloheximide, which blocks cyclins production and progression of extract into mitosis (Gillespie et al, 2012). However, to determine the existence of a mitotic replisome disassembly pathway in *Xenopus* egg extract we needed to allow for this progression. To achieve this, we supplemented the extract with recombinant cyclin after completion of DNA replication. *Xenopus* egg extract synthesises cyclin A1 (embryonic form of cyclin A), B1, and B2 (Minshull et al, 1989). Whereas the B family of cyclins has been shown to drive *Xenopus* meiotic division and oocyte maturation (Hochegger et al, 2001), both cyclin A and B have been shown to promote egg extract transition to mitosis (Strausfeld et al, 1996). We therefore purified His-tagged *X. laevis* cyclin A1 NΔ56 (hereafter: cyclin A1Δ) and added it to the extract upon completion of DNA replication, as described previously, to induce mitotic entry (Strausfeld et al, 1996). The N-terminal deletion to cyclin A1 prevents its degradation and ensures that the extract remains arrested in mitosis, reducing de-synchronisation of our experiments. In all of the experiments described below, we supplemented extract with cyclin A1Δ after completion of DNA replication. As a result, addition of cyclin A1Δ did not stimulate any more DNA synthesis (Fig S1A), but it did lead to progression into mitosis, as evidenced by breakage of the nuclear envelope, condensation of chromatin into chromosomes, and phosphorylation of Serine 10 on histone H3, which coincides with chromosome condensation (Fig S1B and C). Moreover, we could detect chromatin binding of condensin Smc2—another clear sign of the mitosis stage (Fig S1C).

To test if the replisome, which is retained on chromatin in S-phase, can be unloaded as cells enter mitosis, we needed to inhibit S-phase replisome disassembly. To achieve this, a replication reaction was set up in the interphase extract supplemented with Cullin ligase inhibitor MLN4924 to block Cul2^LRR1 activity (Sonneville et al, 2017). Addition of MLN4924 to egg extract did not affect its ability to synthesise DNA as shown previously (Moreno

et al, 2014) and in Fig S1A. Moreover, the timing of replication completion was very reproducible within a batch of extract (Fig S1A). Throughout this article, we confirmed the timing of replication completion for every extract used. To do this, we determined the time point, after addition of sperm DNA, when no more $^{32}$P-labelled dATP was incorporated into DNA. At this point, the components of the replisome were also seen to be unloaded from chromatin in the control samples, but retained on chromatin in those samples supplemented with Cullin ligase inhibitor MLN4924 (Fig S1D). We then optionally added cyclin A1Δ at the replication completion time (usually 90 min after sperm DNA addition), isolated chromatin at different time points during mitosis progression, and analysed chromatin-bound proteins by Western blotting (Fig 1A and B). The presence of the Cullin ligase inhibitor MLN4924 in the S-phase extract did not affect the DNA synthesis level nor induction of mitosis in our extract (Fig S1A and C). As seen in Fig 1B, in control samples without inhibition of replisome disassembly in S-phase, there were no CMG helicase components (hereafter represented by Cdc45 and Psf2 subunits) associated with chromatin at any times analysed, as replisome disassembly takes place before addition of cyclin A1Δ. Notably, we did detect low levels of PCNA bound to chromatin in late S-phase and mitosis as it remains on DNA after Okazaki fragment maturation and completion of replication, so as to aid post-replicative DNA repair (Gao et al, 2017). When replisome disassembly in S-phase was blocked with MLN4924 treatment, the CMG helicase remained associated with chromatin, as expected, and Mcm7 displayed low levels of ubiquitylation. Similar low levels of Mcm7 ubiquitylation have been shown previously upon both MLN4924 treatment and Cul2 immunodepletion (Moreno et al, 2014; Sonneville et al, 2017) and could indicate residual activity of the Cul2^LRR1 ligase or activity of yet another unidentified ligase. Importantly, upon addition of cyclin A1Δ, Cdc45, Psf2, and ubiquitylated Mcm7 were efficiently unloaded (Fig 1B). This result indicates that indeed the mitotic replisome disassembly pathway is evolutionarily conserved and that, unlike the S-phase pathway, it does not require the activity of Cullin-type ubiquitin ligases because the Cullin ligase inhibitor MLN4924 was present throughout the reaction. The continuous presence of unmodified Mcm7 in our samples is a result of the high quantity of DNA used in our experiments. This allows us to clearly detect the replication fork components and ubiquitylated Mcm7 by Western blotting. Because of the high quantity of DNA used, some of the nuclei were not able to form completely and failed to initiate replication, resulting in the isolation of unfired Mcm2-7 complexes, as shown previously (Moreno et al, 2014). However, when we added a much lower quantity of DNA and used minimal licensing conditions (addition of recombinant geminin, 2 min after sperm DNA), we could detect unloading of Mcm7 in mitosis together with Cdc45 and Psf2 (Fig S2A).

Next, we tested whether the mitotic replisome disassembly pathway requires the activity of the p97 segregase. We followed the experimental setup as before but now optionally added the inhibitor of p97, NMS873, along with cyclin A1Δ to inhibit p97 activity during mitosis. Addition of NMS873 to the extract together with cyclin A1Δ did not affect extract transition to mitosis (Fig S2B). In these conditions, the retained replisome was unloaded upon cyclin A1Δ addition in the absence but not in the presence of the p97

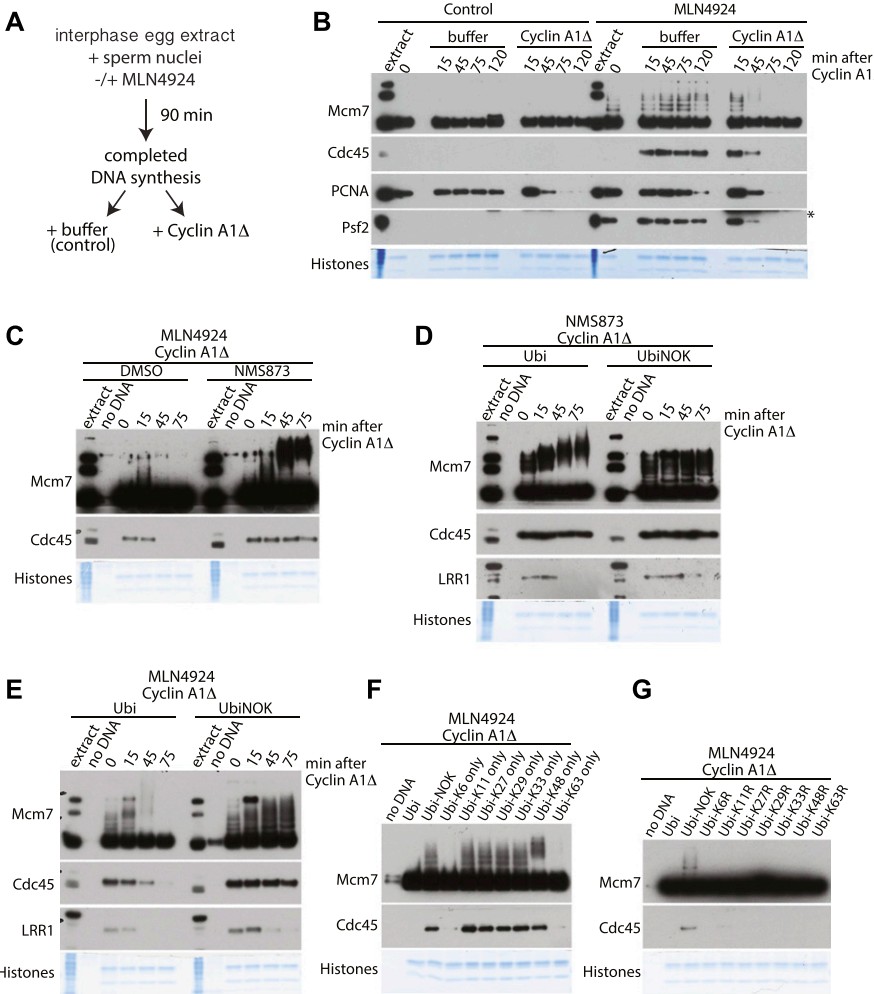

**Figure 1. Mcm7 is ubiquitylated with K6 and K63 ubiquitin chains in mitosis and removed from chromatin by p97 segregase.**

**(A)** Experimental design for driving egg extract into mitosis. **(B)** Experiment following design in (A). DNA was replicated to completion (90 min) in egg extract supplemented optionally with a Cullin ligase inhibitor MLN4924. After completion of the replication reaction, cyclin A1Δ was optionally added to the extract to drive extract into mitosis. Chromatin was isolated at indicated time points after cyclin A1Δ addition and chromatin samples analysed by Western blotting with the indicated antibodies. Time "0" sample was isolated at the replication completion time when cyclin A1Δ was added to the extract. Colloidal Coomassie-stained histones serve as a quality and loading control. An asterisk "*" by the Psf2 blot indicates a band of cyclin A1Δ that is recognised by Psf2 antibody. **(C)** The replication reaction was completed in the presence of Cullin ligase inhibitor MLN4924 and driven into mitosis by addition of cyclin A1Δ. At the same time as cyclin A1Δ, half of the sample was supplemented additionally with p97 inhibitor NMS873. Chromatin samples were isolated at indicated time points and analysed as in (B). A sample without DNA addition (no DNA) was processed alongside others as a chromatin specificity control. **(D)** The replication reaction was completed in the presence of p97 inhibitor NMS873 and driven into mitosis by addition of cyclin A1Δ. At the same time as cyclin A1Δ, the samples were supplemented with recombinant wt ubiquitin or UbiNOK. Chromatin samples were analysed as above. **(E)** Experiment as in (D) but replication reaction was carried out in the presence of Cullin ligase inhibitor MLN4924 instead of p97 inhibitor NMS873. **(F, G)** Replication reaction was completed in the presence of Cullin ligase inhibitor MLN4924 and driven into mitosis by addition of cyclin A1Δ. At the same time as cyclin A1Δ addition, the extract was supplemented with the indicated mutants of ubiquitin. Chromatin was isolated at 75 min after cyclin A1Δ addition and analysed by Western blotting as above.
Source data are available for this figure.

inhibitor, indicating that indeed p97 does play an essential role in promoting mitotic replisome disassembly (Fig 1C). We could also see an analogous result if the p97 inhibitor was present throughout the two stages of the cell cycle as the only way to block replisome disassembly (Fig S2C). Importantly, the presence of p97 inhibitor NMS873 throughout interphase does not impede DNA synthesis (Sonneville et al, 2017) and does not induce additional DNA synthesis when combined with addition of cyclin A1Δ (Fig S2D).

Interestingly, when mitotic unloading of replisome was blocked with p97 inhibitor, we could clearly see accumulation of highly modified forms of Mcm7 on chromatin (Figs 1C and S2C). To examine whether these modifications were due to further ubiquitylation of Mcm7 in mitosis, we blocked S-phase and mitotic replisome disassembly by addition of p97 inhibitor from the beginning of the replication reaction, induced mitosis after completion of DNA synthesis, and optionally supplemented extract with a high concentration of wt ubiquitin (Ubi) or a chain-terminating mutant of ubiquitin with all lysines mutated (UbiNOK). Supplementation of extract with UbiNOK at the same time as cyclin A1Δ addition allowed for normal entry into mitosis as indicated by nuclear envelope breakdown (Fig S2E). Addition of Ubi allowed for accumulation of highly modified Mcm7 on chromatin in mitosis as before (Fig 1D), but

UbiNOK blocked further modifications of Mcm7, leaving only the chains which were built previously in S-phase (Fig 1D). To determine whether this further Mcm7 polyubiquitylation in mitosis is essential for mitotic replisome disassembly, we repeated the experiment with addition of wt Ubi or UbiNOK, but this time only in the presence of the Cullin ligase inhibitor MLN4924 from the start of the reaction (Fig 1E). Indeed, addition of UbiNOK to mitotic extract did block disassembly of the replisome (as shown by permanent Cdc45 chromatin binding), suggesting that further Mcm7 polyubiquitylation is required for mitotic replisome unloading. We also observed that LRR1 (the substrate-specific subunit of Cullin 2, targeting Mcm7 in S-phase) dissociates from chromatin in mitosis irrespectively of replisome disassembly, in agreement with the finding that it does not play an essential role in this pathway (Fig 1D and E). Importantly, these results indicate that a previously unreported ubiquitin ligase is needed for Mcm7 ubiquitylation and replisome disassembly in mitosis.

As the ubiquitin ligase acting in the mitotic pathway differed from that of the S-phase pathway, we decided to test whether the type of ubiquitin chains built on Mcm7 in mitosis also differed. To determine which ubiquitin chains are required for mitotic Mcm7 ubiquitylation and replisome disassembly, we supplemented

extract with Cullin ligase inhibitor MLN4924, allowed for completion of DNA synthesis, and subsequently induced mitosis along with addition of a series of ubiquitin mutants that have only one lysine left in their sequence (Fig 1F). We observed that only wt ubiquitin and ubiquitin containing only lysine 6 (K6 only) or lysine 63 (K63 only) could support mitotic replisome disassembly (as visualised by the absence of Cdc45 on chromatin at 75 min after inducing mitosis) (Fig 1F). Interestingly, chains linked through lysine 48 (K48), which are responsible for S-phase unloading (Moreno et al, 2014), could still be attached to Mcm7 in mitosis (upshift of modified Mcm7 forms), but they could not support unloading of the replisome as Cdc45 remained associated with chromatin. In a reciprocal experiment, we used a series of ubiquitin mutants with only one of the lysines within ubiquitin mutated (Fig 1G). All of the mutants used, apart from the UbiNOK control mutant, supported disassembly of the replisome, suggesting that either K6 or K63 can fulfil the mitotic pathway requirements (Fig 1G).

Having established that the type of ubiquitin chains and the type of ubiquitin ligase used by the mitotic pathway of replisome disassembly were different to those acting in the S-phase pathway, our aim was to identify this ubiquitin ligase. To this end, we decided to immunoprecipitate the replisome retained on mitotic chromatin and analyse all the interacting proteins by mass spectrometry. We set up a replication reaction in the presence of caffeine and the p97 inhibitor NMS873 and induced mitosis upon completion of DNA synthesis (90 min). Neither of the treatments affected the extract's ability to synthesise DNA (Fig S3A). We then immunoprecipitated Mcm3 from mitotic chromatin and analysed the interacting factors by mass spectrometry. First, we determined which components of the replisome are still retained on chromatin in mitosis. For this, we compared the replisome components retained on chromatin in mitosis with S-phase post-termination replisome, reported previously (Sonneville et al, 2017) (Fig 2A and B). Interestingly, although inhibition of replisome disassembly in the S-phase led to accumulation of the whole replisome on chromatin (Sonneville et al, 2017), only a selection of replisome components stayed on chromatin in mitosis. All of the lagging strand components of the replisome were lost, as were Mcm10 and Claspin, whereas levels of Ctf4/And-1, Timeless, Tipin, and Pol epsilon were also reduced (Fig 2A and B). This suggests that only components directly interacting with the CMG remained accumulated around it through to mitosis, whereas others, more peripheral to CMG, could dissociate over time.

The level of histone chaperone FACT (Spt16 and SSRP) stayed the same between S-phase and mitosis. This suggests that the retained replisome in mitosis has the potential ability to move through chromatin as FACT is likely to displace nucleosomes in front of such a replisome. We could see also that Cul2$^{LRR1}$, which strongly accumulated in the S-phase post-termination replisome, is not a major component of the mitotic replisome, as expected from previous data (Fig 1D and E).

Finally, we detected two other ubiquitin ligases interacting with the mitotic helicase: TRAIP and RNF213. More specifically, we found that TRAIP interacts with the post-termination replisome in S-phase, but it is enriched in mitosis, whereas RNF213 is a minor interactor of only the mitotic replisome (Fig 2A). The TNF-receptor–associated factor (TRAF)–interacting protein (TRAIP, also

known as TRIP or RNF206) was originally identified through its ability to bind TRAF1 and TRAF2 and shown to inhibit NFkB activation (Lee et al, 1997). It has been since shown that TRAIP is an E3 ubiquitin ligase, which is essential for cell proliferation (Besse et al, 2007; Park et al, 2007), and which is required for resolution of replication stress (Feng et al, 2016; Harley et al, 2016; Hoffmann et al, 2016) and for regulation of the spindle assembly checkpoint during mitosis (Chapard et al, 2014). TRAIP is ubiquitously expressed, with its expression regulated by E2F transcription factors and protein stability controlled by the ubiquitin proteasome pathway—as a result, the protein level of TRAIP peaks in the G2/M stage of the cell cycle (Chapard et al, 2015). On the other hand, RNF213 (mysterin) is a large (591 kD) ATPase/E3 ligase, which is mostly known as being a susceptibility gene for moyamoya disease (cerebrovascular disease) (Kamada et al, 2011; Liu et al, 2011). Of note, $RNF213^{-/-}$ mice do not show any apparent health problems (Kobayashi et al, 2013; Sonobe et al, 2014) and more recently, RNF213 was shown to globally regulate ($\alpha$-ketoglutarate)-dependent dioxygenases ($\alpha$-KGDDs) and non-mitochondrial oxygen consumption (Banh et al, 2016). To support our mass spectrometry data, we tested a number of antibodies by Western blotting against RNF213 and TRAIP to confirm their association with the chromatin-bound replisome in mitosis. Although we were unsuccessful with detection of any signal for RNF213, we were able to show that TRAIP interacts with the replisome retained on chromatin in mitosis (Figs 2C and S3B).

After confirming that the ubiquitin ligase TRAIP is the likely candidate responsible for Mcm7 ubiquitylation and replisome disassembly in mitosis, we next characterised TRAIP chromatin binding dynamics during the two cell cycle stages and the replisome disassembly process. We found that although TRAIP associated weakly with the S-phase chromatin at times when forks progress through chromatin and replicate DNA, it accumulated strongly on the S-phase chromatin upon inhibition of replisome disassembly with the p97 inhibitor NMS873 (Fig 3A). Importantly, TRAIP also accumulated on mitotic chromatin when replisome disassembly was inhibited with the p97 inhibitor, following the same pattern as replisome components (Fig 3B). To test whether TRAIP is indeed the ubiquitin ligase responsible for unloading of replisome in mitosis, we aimed to inhibit TRAIP enzymatic activity in our extract. As we were unable to efficiently immunodeplete TRAIP from the egg extract with any of the antibodies tested, we decided to use a dominant-negative, ligase-dead mutant of TRAIP to out-compete the endogenous TRAIP. To this end, we purified recombinant His/SUMO-tagged X. laevis TRAIP, both wt and the C25A RING domain mutant, which has been shown to disrupt TRAIP ubiquitin ligase activity (Besse et al, 2007; Chapard et al, 2014) and (Fig S3C). We blocked disassembly of the replisome in S-phase by addition of the Cullin ligase inhibitor MLN4924 and drove extract into mitosis by addition of cyclin A1Δ, when we added recombinant wt or mutant TRAIP. Addition of neither wt nor mutant TRAIP affected the extract's ability to enter mitosis upon cyclin A1Δ addition, as shown through nuclear envelope breakdown (Fig S3D) and Smc2 chromatin loading (Fig 3C and D). As shown in Fig 3C, addition of the enzymatic dead mutant of TRAIP into mitotic extract inhibits unloading of post-termination replisomes retained on chromatin, whereas addition of wt TRAIP does not have such an effect. To confirm that enzymatic dead TRAIP indeed affects replisome

**A**

| | Protein (kDa) | IP from mitotic chromatin with retained CMG (extract treated with p97i ) | | IP from S-phase post-termination chromatin with retained replisome (Sonneville et al 2017) | |
|---|---|---|---|---|---|
| | | IgG TSC (coverage) | α-Mcm3 TSC (coverage) | IgG TSC (coverage) | α-Mcm3 TSC (coverage) |
| **CMG complex (Mcm2-7/Cdc45/GINS)** | Mcm2 (100) | 364 (46%) | 1348 (78%) | 17 (17%) | 975 (71%) |
| | Mcm3 (90) | 222 (55%) | 1610 (90%) | 17 (20%) | 1111 (84%) |
| | Mcm4 (97) | 198 (45%) | 1277 (85%) | 7 (7.4%) | 947 (75%) |
| | Mcm5 (82) | 70 (59%) | 720 (88%) | 30 (17%) | 927 (83%) |
| | Mcm6 (93) | 196 (60%) | 964 (91%) | 39 (14%) | 888 (83%) |
| | Mcm7 (82) | 160 (52%) | 1082 (84%) | 24 (8.9%) | 905 (73%) |
| | Cdc45 (66) | 18 (26%) | 187 (46%) | 2 (2.6%) | 277 (54%) |
| | Psf1 (23) | 3 (31%) | 58 (90%) | 0 | 62 (85%) |
| | Psf2 (21) | 0 | 26 (58%) | 0 | 33 (79%) |
| | Psf3 (24) | 3 (15%) | 55 (79%) | 0 | 55 (92%) |
| | Sld5 (26) | 2 (10%) | 42 (81%) | 0 | 44 (64%) |
| **Replisome components** | Spt16 (118) | 109 (23%) | 431 (61%) | 3 (2.9%) | 453 (52%) |
| | Ssrp1 (79) | 19 (8%) | 224 (45%) | 0 | 229 (48%) |
| | Ctf4 (97) | 4 (4%) | 82 (31%) | 0 | 166 (49%) |
| | Timeless (149) | 6 (3%) | 112 (39%) | 0 | 171 (29%) |
| | Tipin (40) | 0 | 20 (27%) | 0 | 54 (24%) |
| | Claspin (146) | 0 | 0 | 0 | 72 (21%) |
| | Mcm10 (95) | 0 | 0 | 0 | 257 (44%) |
| | Ctf18 (113) | 0 | 0 | 0 | 172 (36%) |
| | Ctf8 (14) | 0 | 0 | 0 | 8 (52%) |
| | Dcc1 (45) | 0 | 0 | 0 | 33 (38%) |
| | Rfc2 (38) | 0 | 0 | 0 | 27 (54%) |
| | Rfc3 (40) | 0 | 0 | 4 (14%) | 36 (54%) |
| | Rfc4 (40) | 0 | 0 | 2 (5.2%) | 31 (57%) |
| | Rfc5 (38) | 0 | 0 | 0 | 22 (56%) |
| | Polα1 (165) | 0 | 0 | 0 | 109 (30%) |
| | Prim1 (49) | 0 | 0 | 0 | 14 (28%) |
| | Prim2 (61) | 0 | 0 | 0 | 24 (24%) |
| | Polε1 (261) | 12 (5%) | 166 (29%) | 2 (0.88%) | 543 (38%) |
| | Polε2 (60) | 0 | 15 (32%) | 0 | 103 (50%) |
| | Polε3 (17) | 0 | 4 (27%) | 0 | 10 (45%) |
| | Polε4 (12) | 0 | 0 | 0 | 4 (35%) |
| | Rpa1 (67) | 80 (53%) | 107 (48%) | 0 | 10 (12%) |
| | Rpa2 (29) | 13 (22%) | 30 (55%) | 0 | 2 (9%) |
| | Rpa3 (13) | 6 (52%) | 10 (92%) | 0 | 0 |
| **cullin2-LRR1** | cullin2 (87) | 6 (8%) | 4 (3%) | 0 | 124 (52%) |
| | LRR1 (47) | 0 | 11 (21%) | 0 | 46 (44%) |
| | Elongin B (13) | 0 | 0 | 0 | 5 (31%) |
| | Elongin C (12) | 0 | 0 | 0 | 7 (32%) |
| **ubi ligases** | Traip (87) | 2 (4%) | 42 (44%) | 0 | 34 (36%) |
| | RNF213 (47) | 0 | 17 (2%) | 0 | 0 |

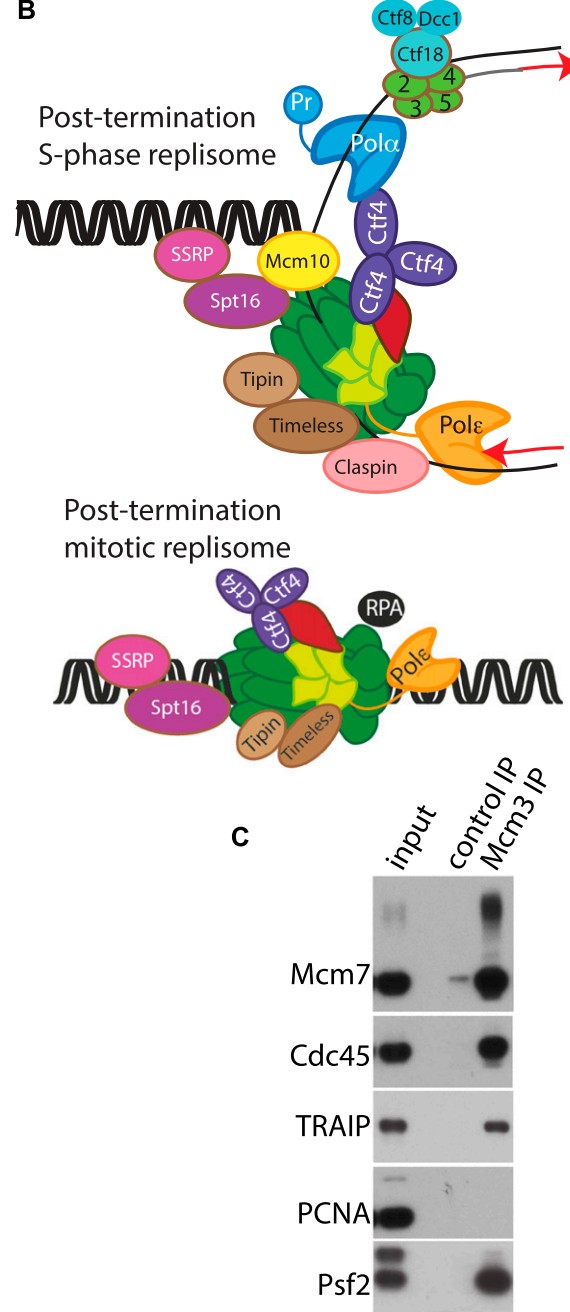

**B**

**C**

**Figure 2. Composition of the replisome retained on mitotic chromatin.**
**(A)** The replication reaction was completed in egg extract supplemented with caffeine and p97 inhibitor NMS873. The extract was then driven into mitosis by addition of cyclin A1Δ. Chromatin was isolated at 60 min after cyclin A1Δ addition and chromatin proteins released from DNA. The DNA synthesis kinetics are provided in Fig S3A. Antibodies against Mcm3 (or control IgG) were used to immunoprecipitate replisomes, and the immunoprecipitated samples were analysed by mass spectrometry. The total spectral count for each identified replisome component is presented together with sequence coverage of analysed peptides. The results for this analysis of mitotic retained replisome are compared with the S-phase post-replication replisome reported in Sonneville et al (2017). **(B)** Schematic representation of the data presented in (A). **(C)** A small proportion of the material from the mitotic Mcm3 IP experiment in (A) was analysed by Western blotting with indicated antibodies.

unloading through ubiquitylation of Mcm7 in mitosis, we repeated this experiment but supplemented the mitotic extract also with p97 inhibitor NMS873 to inhibit unloading of ubiquitylated Mcm7. Fig 3D shows that addition of enzymatic dead TRAIP, but not wt TRAIP, perturbs mitotic ubiquitylation of Mcm7 as the ubiquitylated forms of Mcm7 remain very close in size to the chains built on Mcm7 already in S-phase (15-min time point). The same prevention of ubiquitylation of Mcm7 in mitosis was observed when we used recombinant GST-tagged TRAIP C25A mutant, but not wt GST-TRAIP (Fig S3F). These results suggest that the recombinant mutant of

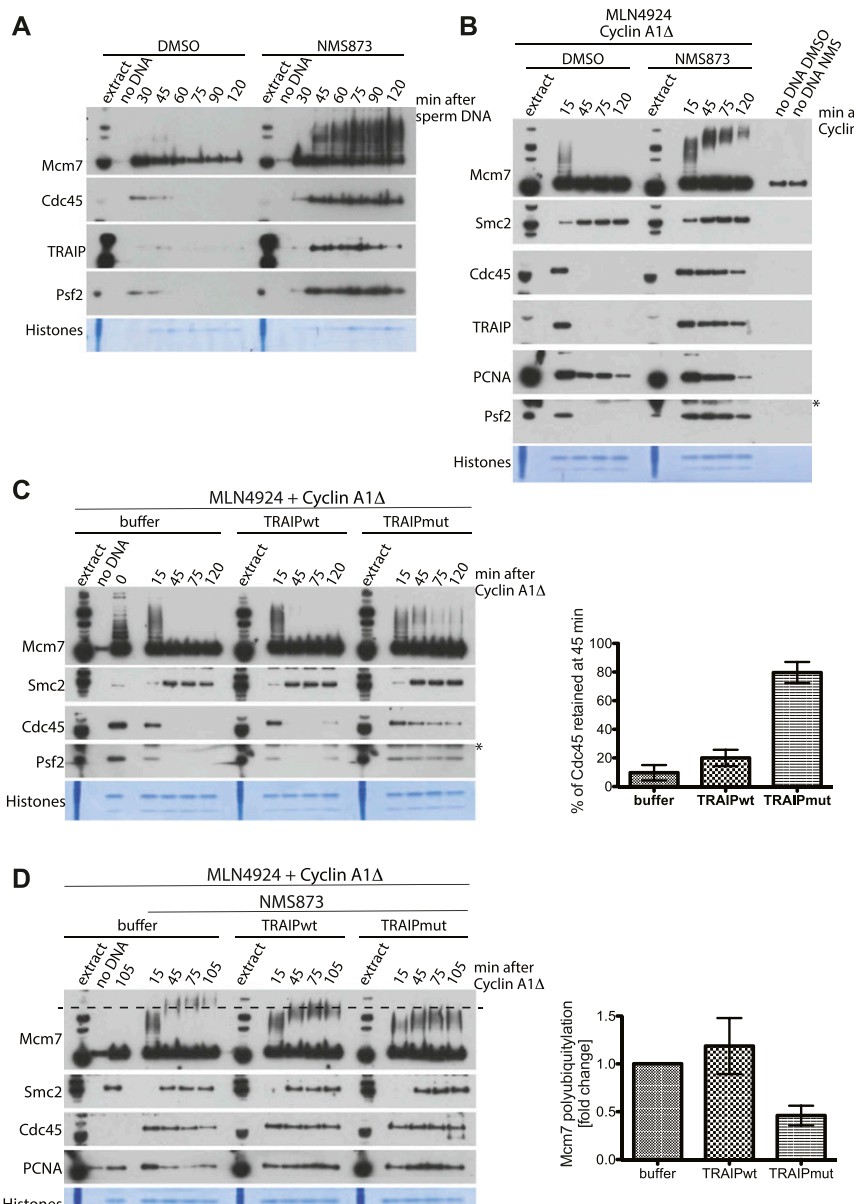

**Figure 3. TRAIP ubiquitin ligase drives replisome disassembly in mitosis.**
**(A)** Sperm DNA was replicated in egg extract optionally supplemented with p97 inhibitor NMS873. Chromatin samples were isolated during the reaction at indicated time points and analysed as in Fig 1. Colloidal Coomassie-stained histones serve as a quality and loading control. A sample without DNA addition (no DNA) was processed alongside others as a chromatin specificity control. **(B)** Experiment analogous to Fig 1C but analysed with indicated antibodies. An asterisk "*" by the Psf2 blot indicates the band of cyclin A1Δ that is recognised by the Psf2 antibody. **(C)** (left) - The replication reaction was completed in the presence of Cullin ligase inhibitor MLN4924 and driven into mitosis by addition of cyclin A1Δ. At the same time as cyclin A1Δ, the samples were supplemented optionally with LFB1/50 buffer, wt His/SUMO–tagged TRAIP, or RING-mutant (C25A) TRAIP to a final concentration of 50 µg/ml. Chromatin samples were isolated at indicated time points and analysed with indicated antibodies. Time "0" sample was isolated at the replication completion time when cyclin A1Δ and recombinant TRAIP were added to the extract. An asterisk "*" by the Psf2 blot indicates the band of cyclin A1Δ that is recognised by the Psf2 antibody. (right)- The level of retained Cdc45 on chromatin was quantified at 15 and 45 min in each condition and the percentage of the 15 min signal still retained on chromatin at 45 min calculated. The graph represents a mean of three independent experiments with SEM. **(D)** (left) - The experiment was performed as in (C) but with addition of p97 inhibitor NMS873 at the same time as cyclin A1Δ to block ubiquitylated Mcm7 on chromatin. His/SUMO-tagged TRAIPwt and ligase dead mutant were added to a final concentration of 100 µg/ml. The sample isolated at 105 min without NMS873 provides a control for the unloading without p97 inhibition. The dashed line on the Mcm7 blot runs through the middle of the ubiquitylation signal for Mcm7 in mitosis in the control (buffer) sample to aid comparison of chain lengths between samples. (right) - The Mcm7 polyubiquitylation signal was quantified for each condition as explained in materials and methods and an example provided in Fig S3E. The graph presented here shows the mean fold change of Mcm7 polyubiquitylation signal at 45 min after cyclin A1Δ addition over three independent experiments with SEM.

TRAIP successfully competed with endogenous TRAIP protein and that ubiquitin ligase activity of TRAIP is needed for Mcm7 ubiquitylation and disassembly of post-termination replisome in mitosis. The low level of Mcm7 ubiquitylation visible in samples supplemented with TRAIP mutant is most likely due to the fact that there is still endogenous active TRAIP in the extract.

To fully understand the requirement for ubiquitin-like modifications during mitotic replisome disassembly in vertebrates, we aimed to establish whether SUMOylation plays any role in this process as ULP-4 is essential for mitotic helicase disassembly in *C. elegans* embryos. To this end, we decided to inhibit or stimulate SUMOylation during mitosis and assess its effect on replisome disassembly. First, we observed that the late S-phase chromatin is full of SUMO2ylated factors and that levels of these proteins go down over time upon

entry into mitosis (Fig 4). To inhibit SUMOylation, we supplemented the mitotic extract with the recombinant active domain of SENP1, which acts as a potent nonspecific deSUMOylating enzyme. Addition of SENP1 indeed wiped out all the SUMO2ylation (Fig 4A) and SUMO1ylation (Fig S5A), but disassembly of the mitotic replisome is not affected (Fig 4A). We also stimulated SUMOylation through addition of a high concentration of recombinant SUMO1 or SUMO2 (Fig S4). In both cases, despite a clear increase of SUMO signal on chromatin, unloading of the mitotic replisome was not affected. Finally, we also blocked de-SUMOylation with SUMO2-VS, a derivative of SUMO2, which binds to the active site of SENPs and blocks their activity. Again, we observed strong accumulation of SUMO2ylated products in the extract and on chromatin without affecting mitotic replisome disassembly (Fig 4B). Interestingly, despite inhibition of

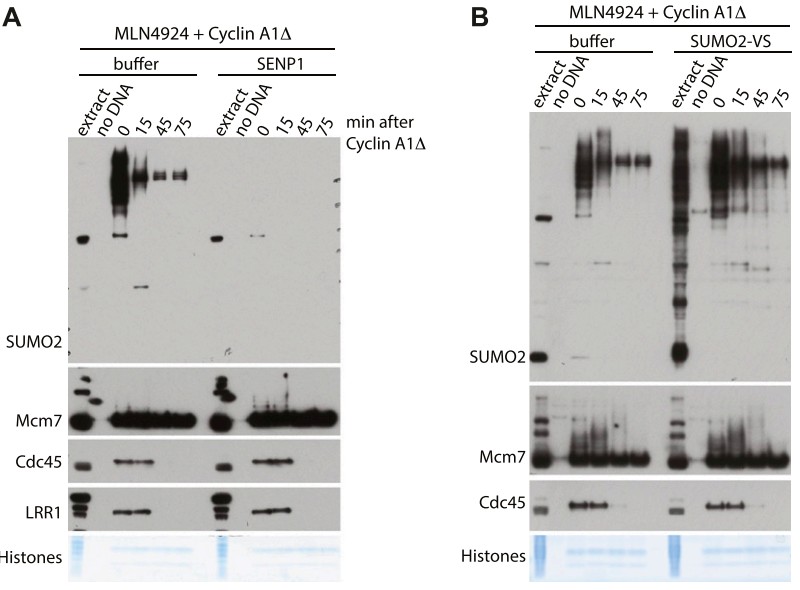

**Figure 4. SUMOylation is not required for mitotic replisome disassembly.**
**(A)** The replication reaction was completed in the presence of Cullin ligase inhibitor MLN4924 and driven into mitosis by addition of cyclin A1Δ. At the same time as cyclin A1Δ, half of the sample was supplemented additionally with the active domain of SENP1. Chromatin samples were isolated at indicated time points and analysed as in Fig 1. Colloidal Coomassie-stained histones serve as a quality and loading control. A sample without DNA addition (no DNA) was processed alongside others as a chromatin specificity control. **(B)** As in (A) but instead of supplementing extract with SENP1, it was supplemented with SENPs inhibitor SUMO2-VS.

de-SUMOylating enzymes, most of the SUMO signal is still disappearing from chromatin during progression of mitosis, indicating that the SUMOylated proteins are unloaded from chromatin throughout mitosis rather than being de-SUMOylated. In conclusion, we determined that SUMO modifications do not play an essential role in the mitotic replisome disassembly pathway in *Xenopus* egg extract. In an analogous way, we have also shown that they do not play a role during the S-phase replisome disassembly pathway (Fig S5A and B).

Finally, we set out to determine whether this mitotic replisome disassembly pathway was a mere "backup" pathway for replisomes that terminated in S-phase but failed to be unloaded, or if it has a more generic ability to remove any replication machinery still remaining on chromatin in mitosis. To test such a possibility, we stalled replisomes on chromatin by addition of DNA polymerase inhibitor aphidicolin to the egg extract during the DNA replication reaction. To accumulate such replisomes in large numbers, we also supplemented the extract with caffeine so as to block checkpoint activation and fire origins uncontrollably. Upon accumulation of such blocked replisomes, we supplemented the reaction optionally with cyclin A1Δ at 90 min to induce mitotic entry (Fig 5A). Addition of cyclin A1Δ did not stimulate any more DNA synthesis in our samples (Fig S6). Interestingly, active replisomes remained associated with chromatin throughout the experiment in late S-phase (buffer), with no indication of Mcm7 ubiquitylation as expected (Moreno et al, 2014). Upon addition of cyclin A1Δ, however, Mcm7 became ubiquitylated and replisomes were unloaded (Fig 5A, cyclin A1Δ), although we did observe a slight delay in both of these processes compared with terminated replisomes (compare Fig 5A with Fig 1B). Such a delay is likely due to the fact that with no prior ubiquitylation of Mcm7 in S-phase, it takes longer for ubiquitin chains to be built in mitosis. We also determined that unloading of stalled replisomes requires the activity of p97 segregase, as unloading is inhibited in the presence of the p97 inhibitor NMS873 (Fig 5B). Finally, to test whether the activity of TRAIP ubiquitin ligase is needed for the

unloading of stalled replisomes, we added recombinant wt or enzymatic dead TRAIP to mitotic extract and observed its effect on unloading of such stalled replisomes. As with post-termination replisomes (Fig 3C and D), the enzymatic dead mutant of TRAIP inhibited unloading of stalled helicase (Fig 5C), whereas we also observed a clear reduction in the ubiquitylation of stalled Mcm7 (Fig 5D). From these observations, we can thus say that neither prior modification of Mcm7 in S-phase nor the "terminated" conformation of the helicase are essential for mitotic modification of Mcm7 by TRAIP and subsequent replisome disassembly.

## Discussion

We have presented here the existence of a mitotic pathway of replisome disassembly in *X. laevis* egg extract. One immediate question is why would the cells need a mitotic pathway of replisome disassembly? Traditionally, it is perceived that all DNA metabolism should be finished before cells enter mitosis. According to this model, the G2 phase of the cell cycle is there to ensure that all DNA replication and damage repair are completed before chromosome condensation and segregation during mitosis. The last decade provided, however, much evidence that this is not the case: unreplicated DNA is detected in many human cells in mitosis; DNA synthesis can proceed during mitosis (mitotic DNA synthesis—MiDAS); under-replicated DNA can lead to the formation of ultrafine bridges in anaphase and structures in the G1 stage of the next cell cycle that are bound by 53BP1 protein (53BP1 bodies) (Liu et al, 2014; Minocherhomji et al, 2015; Moreno et al, 2016). Genome-wide, such unreplicated regions correlate with common fragile sites, which are chromosomal loci responsible for the majority of the rearrangements found in cancer cells (Bhowmick & Hickson, 2017). These unreplicated fragments of DNA result from replication forks not finishing replication and such forks, with their associated replisomes,

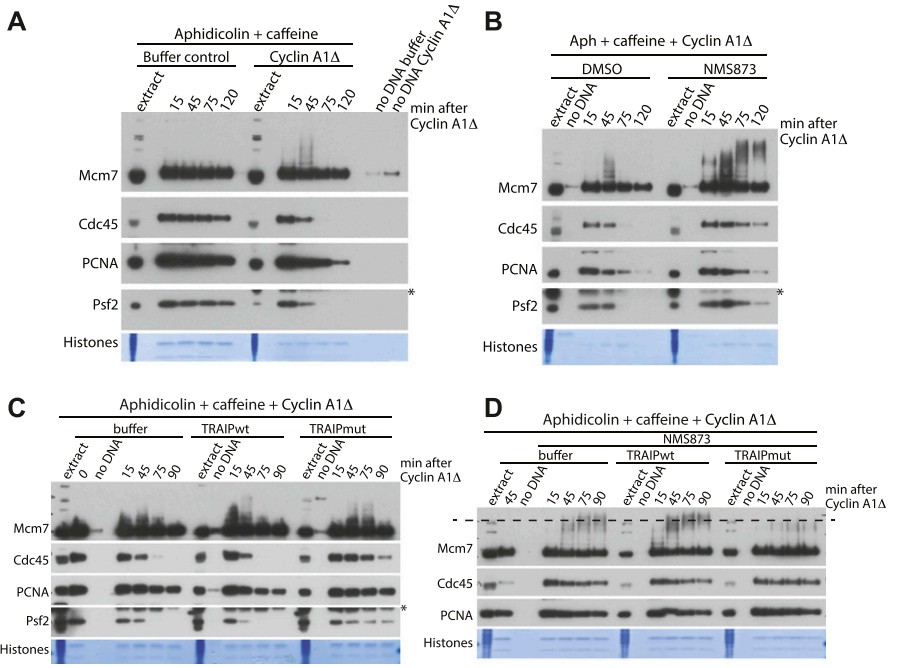

**Figure 5. Mitotic unloading of stalled helicases.**
**(A)** The replication reaction was performed in egg extract supplemented with DNA polymerase inhibitor aphidicolin and checkpoint inhibitor caffeine. After 90 min of reaction, cyclin A1Δ was optionally added, and the chromatin samples were isolated during the reaction at indicated time points and analysed as in Fig 1. Colloidal Coomassie-stained histones serve as a quality and loading control. A sample without DNA addition (no DNA) was processed alongside others as a chromatin specificity control. An asterisk "*" by the Psf2 blot indicates the band of cyclin A1Δ that is recognised by the Psf2 antibody. **(B)** The inhibition of stalled replisomes was achieved as in (A), and the extract was driven into mitosis by addition of cyclin A1Δ and optional supplementation with p97 inhibitor NMS873. The chromatin samples were analysed as in (A). **(C)** Replication reaction was performed with addition of aphidicolin and caffeine, at 90 min, cyclin A1Δ was added and optionally: buffer, His/SUMO-TRAIPwt, or ligase dead His/SUMO-TRAIPmut (C25A) to a final concentration of 50 μg/ml. The chromatin samples were isolated at indicated time points and analysed with indicated antibodies. Time "0" sample was isolated at the replication completion time when cyclin A1Δ and recombinant TRAIP were added to the extract. **(D)** The experiment was performed as in (C) but with addition of p97 inhibitor NMS873 at the same time as cyclin A1Δ to accumulate ubiquitylated forms of Mcm7 on chromatin. His/SUMO–tagged TRAIPwt and mutant were added to a final concentration of 100 μg/ml. The sample isolated at 45 min without NMS873 provides a control for the replisome unloading without p97 inhibition. The dashed line on the Mcm7 blot runs through the middle of the ubiquitylation signal for Mcm7 in mitosis in the control (buffer) sample to aid comparison of chain lengths between samples.

are subsequently retained on chromatin into mitosis. It is likely that these unreplicated DNA fragments must be processed in mitosis to ensure correct chromosome segregation, and this processing will involve replisome unloading and fork remodelling —hence the need for a process of replisome disassembly in mitosis.

TRAIP is a pleiotropic ubiquitin ligase involved in numerous cellular processes. It is clear that TRAIP is essential for appropriate repair of DNA damage in many forms: mitomycin C–induced inter-strand crosslinks; damage caused by treatments with camptothecin (Hoffmann et al, 2016), UV (Harley et al, 2016) and hydroxyurea (Feng et al, 2016); as well as for translesion DNA synthesis (Wallace et al, 2014). TRAIP has also been reported to be an important regulator of the spindle assembly checkpoint and regulates mitotic progression (Chapard et al, 2014; Park et al, 2015). For most of these processes, the ubiquitin ligase activity of TRAIP is essential, but the substrate (s) modified by TRAIP is not known.

In support of our observation that TRAIP interacts weakly with the S-phase chromatin when replication forks replicate DNA (Fig 3A), TRAIP has been shown to interact with nascent DNA in unperturbed S-phase in human cells through nascent chromatin capture (Hoffmann et al, 2016), but TRAIP knockdown does not significantly affect replication progression and overall DNA synthesis rates (Harley et al, 2016; Hoffmann et al, 2016). Upon DNA damage, TRAIP relocalises from nucleoli to sites of damage in a manner dependent on a PCNA interacting box (PIP-box), present at the C terminus of TRAIP (Feng et al, 2016; Hoffmann et al, 2016). Loss of TRAIP was suggested to interfere with the reconfiguration of stalled replication forks (possibly through unloading of PCNA) (Hoffmann et al, 2016), as further inhibition of proteasomal degradation in the absence of TRAIP did not exacerbate the levels of hydroxyurea-induced fork stalling. This suggested that degradation of a TRAIP ubiquitylation substrate is not the cause of this phenotype (Feng et al, 2016; Hoffmann et al, 2016). Interestingly, cells expressing the ΔRING mutant of TRAIP as the only TRAIP version, are as sensitive to mitomycin C as TRAIP knockdown cells, while ΔPIP TRAIP cells are only mildly sensitive. This indicates that even without PCNA interaction, TRAIP can still find its targets at the replication forks (Hoffmann et al, 2016). With the data presented here, identifying TRAIP as the ubiquitin ligase needed for Mcm7 ubiquitylation during mitosis, it is interesting to speculate that TRAIP can play an analogous role during DNA damage repair, that is, to stimulate replisome unloading and fork remodeling. Indeed, recently TRAIP has been shown to ubiquitylate CMG during inter-strand crosslink (ICL) repair for replisome unloading (Wu et al, 2019).

Our data are consistent with a model in which TRAIP drives mitotic replisome disassembly by promoting Mcm7 modification with K6- and K63-linked ubiquitin chains. Although there is no previous experimental evidence that TRAIP can support such ubiquitin linkages in vivo, in vitro assays have shown that TRAIP works well with conjugating enzymes (E2s) UbcH5a,b, and c (but not UbcH2, H3, H6, H7, or Ubc13+Uev1A) (Besse et al, 2007) and (Fig S3C). Interestingly, UbcH5a was shown to support formation of ubiquitin chains with no specific topology (Windheim et al, 2008). It is, therefore, plausible that TRAIP/UbcH5 can effectively produce chains of different linkages to support mitotic replisome disassembly.

Replisome disassembly in S-phase is driven by Mcm7 ubiquitylation, specifically with K48-linked ubiquitin chains. In mitosis, however, K48-linked chains are not functional and unloading is driven instead by K6- and K63-linked chains. We know that p97, in complex with Ufd1 and Npl4 cofactors, is responsible for

unloading of the replisome in S-phase (Moreno et al, 2014; Maric et al, 2017; Sonneville et al, 2017). Although p97 is well known for processing substrates ubiquitylated with K48-linked ubiquitin chains (Meyer et al, 2012), less is known about its contribution in processing other ubiquitin linkages. Interestingly, a recent study shows that upon inhibition of p97 activity, human cells accumulate K6-, K11-, K48- and, to a lesser extent, K63-linked ubiquitin chains (Heidelberger et al, 2018). Moreover, out of five tested p97 cofactors, all were found to associate with K11 chains, four with K48 chains, and three with K63 chains (Alexandru et al, 2008). p97 cofactors are known also to interact with ubiquitin-like modifiers, for example, Nedd8 and Atg8 (reviewed in Meyer (2012)). Finally, p97 was also shown to bind more readily to branched K11-K48 chains than to K11 or K48 chains on their own (Meyer & Rape, 2014). These proteome-wide data imply that the role of p97 does indeed extend beyond recognition of K48-chain-modified substrates, although currently, little is known about its interaction with K6 chains.

Finally, we have shown that in the *Xenopus* system, neither the S-phase (Fig S5) nor the mitotic replisome disassembly requires SUMO modifications (Figs 4 and S4) in contrast to *C. elegans* embryos where ULP-4 is required for mitotic unloading (Sonneville et al, 2017). This requirement may be specific to worm embryos, require ULP-4 protein but not its enzymatic activity, or it may regulate an indirect process that is not well recapitulated in the egg extract cell-free system. Of note, it has been suggested recently that SUMOylation of TRAIP can regulate its stability and ability to move to the nucleus (Park et al, 2016), but this may not be present in the egg extract.

Perturbations in DNA replication initiation and elongation leading to genomic instability are well linked with genetic disorders and can drive cancer development. The disruption of replisome disassembly is, therefore, highly likely to be detrimental to human health too. Although so far we have no solid data to support this claim, previous studies with TRAIP do suggest this to be the case: homozygous *TRAIP* knockout mouse embryos die shortly after implantation because of proliferation defects (Park et al, 2007); mutations in human TRAIP lead to primordial dwarfism (Harley et al, 2016); overexpression of human TRAIP has been reported in basal cell carcinomas (Almeida et al, 2011) and breast cancer (Yang et al, 2006; Zhou & Geahlen, 2009); and reduced nuclear expression of TRAIP was associated with human lung adenocarcinoma (Soo Lee et al, 2016). The fact that cells have evolved multiple pathways to ensure timely replisome disassembly supports the notion of the vital importance of this process for cell biology, and time will tell whether targeting Mcm7 and replisome disassembly in mitosis is the key mechanism leading to any of these disease phenotypes.

# Materials and Methods

## Inhibitors

Caffeine (C8960; Sigma-Aldrich) was dissolved in water at 100 mM and added to the extract along with demembranated sperm nuclei at 5 mM. MLN4924 (A01139; Active Biochem) was dissolved in DMSO at 20 mM and added to the extract 15 min after addition of sperm nuclei at 10 $\mu$M. NMS873 (17674; Cayman Chemical Company) was dissolved in DMSO at 10 mM and added to the extract 15 min after addition of sperm nuclei at 50 $\mu$M. SUMO2-VS (UL-759) was purchased from Boston Biochem and used at 1 $\mu$M in *X. laevis* egg extract. Aphidicolin was dissolved in DMSO at 8 mM and added to the extract along with demembranated sperm nuclei at 40 $\mu$M.

## Recombinant proteins

Recombinant His-tagged ubiquitin and ubiquitin mutants were purchased from Boston Biochem, dissolved in LFB1/50 (40 mM Hepes/KOH, pH 8.0, 20 mM potassium phosphate, pH 8.0, 50 mM KCl, 2 mM MgCl$_2$, 1 mM EGTA, 10% sucrose wt/vol, 2 mM DTT, 1 $\mu$g/ml aprotinin, 1 $\mu$g/ml leupeptin, and 1 $\mu$g/ml pepstatin) buffer at 10 mg/ml, and used at 0.5 mg/ml in *X. laevis* egg extract.

pET28a-*X.l.*SUMO1 and pET28a-X.l.SUMO2 were purchased from GenScript. Recombinant His-tagged *X. laevis* SUMO1 and SUMO2 were expressed in Rosetta (DE3) pLysS cells over night at 20°C after induction with 1 mM IPTG. The cells were lysed in lysis buffer: 50 mM Tris–HCl, 500 mM NaCl, 10 mM imidazole, 2 mM MgCl$_2$, 0.1 mM PMSF, and 1 $\mu$g/ml of each aprotinin, leupeptin, and pepstatin, pH 7.5. Homogenates were supplemented with 25 U/ml benzonase and incubated at room temperature for 20 min. Homogenates were subsequently spun down at 14,000 $g$ for 30 min at 4°C and supernatants incubated with 2 ml of prewashed Super Ni-NTA Affinity Resin (SUPER-NINA100; Generon) for 2 h with rotation at 4°C. Resins were subsequently washed twice with 50 mM Tris–HCl, 500 mM NaCl, 30 mM imidazole, 0.1 mM PMSF, and 1 $\mu$g/ml of each aprotinin, leupeptin, and pepstatin, pH 7.5. Resin-bound proteins were finally eluted in 1 ml fractions with a solution containing 50 mM Tris–HCl, 150 mM NaCl, 200 mM imidazole, 5 mM $\beta$-mercaptoethanol, 0.1 mM PMSF, and 1 $\mu$g/ml of each aprotinin, leupeptin, and pepstatin, pH 7.5. Fractions containing the highest levels of recombinant SUMO1 or SUMO2 were dialysed into LFB1/50 buffer. Both SUMO1 and SUMO2 were used at 0.5 mg/ml in *X. laevis* egg extract.

pET28a-pHISTEV30a-SENP1(415-649) was a kind gift from Prof Ron Hay's laboratory. Recombinant active domain of human SENP1 (aa 415–647) was expressed and purified as explained above for recombinant SUMOs.

Recombinant His-tagged *X. laevis* cyclin A1 N$\Delta$56 (pET23a-*X.l.* Cyclin A1 N$\Delta$56) was a kind gift from Prof Julian Blow's laboratory (Strausfeld et al, 1996), was expressed in Rosetta (DE3) pLysS cells over night at 15°C after induction with 1 mM IPTG, and subsequently purified as explained above for recombinant SUMOs but using different solutions. Lysis buffer: 50 mM Tris–HCl, 300 mM NaCl, 2 mM MgCl$_2$, 1 mM DTT, 0.1 mM PMSF, and 1 $\mu$g/ml of each aprotinin, leupeptin, and pepstatin, pH 7.4. Washes: Resin was washed twice with lysis buffer on its own and twice again with lysis buffer supplemented with 0.1% Triton X-100. Elution buffer: Lysis buffer supplemented with 10% glycerol and 250 mM imidazole.

*Xenopus TRAIP* was cloned into pGS21 vector, expressed in BL21 (DE3) bacterial strain in Auto Induction Media (AIM) media (Formedium) O/N at 18°C. Pellets were lysed in lysis buffer: 50 mM NaH$_2$PO$_4$, pH 9; 300 mM NaCl; 10% glycerol; 2 mM DTT; 2 mM MgCl$_2$; 0.05% Brij; 0.1 mM PMSF; 1 $\mu$g/ml of each aprotinin, leupeptin, and pepstatin; 1 mg/ml lysozyme; and 25 U/ml benzonase. The protein was purified as above but using Glutathione Sepharose 4B (GE

Healthcare) and eluted with 25 mM glutathione. The protein was then dialysed into LFB1/50 buffer (as above) and concentrated up to 0.3 mg/ml of full-length GST-TRAIP. It was used in the egg extract at a final concentration of 30 $\mu$g/ml. pGS21-TRAIP(C25A) was generated by site-directed mutagenesis and purified in an analogous way.

Recombinant His/SUMO–tagged *X. laevis* wt and mutant TRAIP were expressed in Rosetta (DE3) pLysS cells from pCA528 vector O/N at 20°C in AIM media. After pelleting of the bacterial cultures, the cells were lysed in 50 mM NaH$_2$PO$_4$, 500 mM NaCl, 0.05% Brij, 10% glycerol, 10 mM imidazole, 2 mM MgCl$_2$, 0.1 mM PMSF, and 1 $\mu$g/ml of each aprotinin, leupeptin, and pepstatin, pH 9.0. Homogenates were supplemented with 1 mg/ml lysozyme and 25 U/ml BaseMuncher, incubated at room temperature for 20 min and subsequently sonicated (6 × 30 s), and spun (30 min, 31,000 $g$, 4°C). The resulting supernatant was incubated in 2 ml prewashed Super Ni-NTA Affinity Resin (SUPER-NINA100; Generon) O/N with rotation at 4°C. Resins were washed five times with lysis buffer, with respect to the following alterations: Wash 1: 100 mM NaCl, no imidazole. Wash 2: 100 mM NaCl, 20 mM imidazole. Wash 3: 250 mM NaCl, no imidazole. Wash 4 and 5: 500 mM NaCl, 20 mM imidazole. Each respective wash was supplemented with 0.1 mM PMSF and 1 $\mu$g/ml of each aprotinin, leupeptin, and pepstatin. Resin-bound TRAIP was eluted in 1-ml fractions using elution buffer (50 mM NaH$_2$PO$_4$, 500 mM NaCl, 0.05% Brij, 10% Glycerol, 400 mM imidazole, pH 9.0) supplemented with 0.1 mM PMSF and 1 $\mu$g/ml of each aprotinin, leupeptin, and pepstatin. Those fractions containing the highest quantities of wt or mutant TRAIP were dialysed into LFB1/100 buffer.

### In vitro TRAIP autoubiquitylation reaction

The reaction was set up as previously described (Besse et al, 2007). Briefly, GST-TRAIPwt and GST-TRAIPmut were purified as described above but not eluted from Glutathione Sepharose beads. Both beads were then incubated in 100-$\mu$l reaction for 2 h at 37°C with 20 mM Hepes, pH 7.4, 10 mM MgCl$_2$, 1 mM DTT, 60 $\mu$M His-Ubi (Boston Biotech), 50 nM E1 (UBA1/UBE1; Source BioScience), 850, nM E2 UbcH5a (Source BioScience), 1 mM ATP, 30 $\mu$M creatine phosphate, and 1 U of creatine kinase. After incubation, they were extensively washed, boiled in gel-loading buffer, and run on the gel. The membrane was analysed with $\alpha$-ubiquitin antibody.

### Antibodies

$\alpha$-PCNA (P8825), $\alpha$-His (H1029), and $\alpha$-ubiquitin (P4D1) were purchased from Sigma-Aldrich and $\alpha$-phospho-histone H3 (ser10) (D2C8) was purchased from Cell Signalling Technology. $\alpha$-TRAIP (NBP1-87125) and $\alpha$-RNF213 (NBP1-88466) were purchased from Novus Biologicals. $\alpha$-SUMO2 and $\alpha$-SUMO1 were produced in the laboratory by culturing the hybridoma cell lines SUMO2 (8A2), and SUMO1 (21C7), purchased from Developmental Studies Hybridoma Bank (hybridoma cell culture was performed following the manufacturer's instructions and adding 20 mM L-glutamine to the media). Affinity-purified $\alpha$-Cdc45, $\alpha$-Psf2, and $\alpha$-Sld5 (Gambus et al, 2011); $\alpha$-Mcm3 (Khoudoli et al, 2008); $\alpha$-SMC2 (Gillespie et al, 2007); and $\alpha$-LRR1 (S962D) (Sonneville et al, 2017) were previously described. $\alpha$-Mcm7

was raised in sheep against recombinant *X. laevis* Mcm7, purified from *E. coli*, and affinity-purified in the laboratory.

### DNA staining and microscopy

Interphase *X. laevis* egg extract was supplemented with 10 ng/$\mu$l of demembranated sperm nuclei and incubated at 23°C until completion of DNA replication as described before (Gillespie et al, 2012). Mitosis was optionally driven by addition of 826 nM cyclin A1 N$\Delta$56. At time points −30, 0, 30, and 60 min, after addition of cyclin A1, 10 $\mu$l of the reaction was spotted onto a microscope slide with 10 $\mu$l mix of Hoechst 33258 (5824/50; Tocris Bioscience) and DiI stain (D282; Thermo Fisher Scientific), 1 $\mu$g/ml and 10 $\mu$g/ml final concentration, respectively, and incubated at room temperature for 30 min. Nuclei were viewed as previously described (Strausfeld et al, 1996).

### DNA synthesis assay

The replication reactions were started with the addition of demembranated *Xenopus* sperm DNA to 10 ng/$\mu$l as described before (Gillespie et al, 2012). The synthesis of nascent DNA was measured by quantification of P$^{32}$$\alpha$-dATP incorporation into newly synthesised DNA as described before (Gillespie et al, 2012).

### Chromatin isolation time-course

Interphase *X. laevis* egg extract was supplemented with 10 ng/$\mu$l of demembranated sperm DNA and subjected to indicated treatments. The reaction was incubated at 23°C for 90 min to allow completion of DNA replication as described before (Gillespie et al, 2012), after which mitosis was optionally driven by addition of 826 nM cyclin A1 N$\Delta$56. The extract was then also optionally supplemented with inhibitors or recombinant proteins as indicated. Chromatin was isolated in ANIB100 buffer supplemented with 10 mM 2-chloroacetamide (Millipore) and 5 mM N-ethylmaleimide (Acros Organics) at indicated time points after addition of cyclin A1 N$\Delta$56 as previously described (Gillespie et al, 2012).

For the minimal licensing experiment (Fig S2A), interphase *X. laevis* egg extract was supplemented with 5 ng/$\mu$l of demembranated sperm DNA. To minimally license chromatin, Cdt1 activity was blocked through addition of geminin$^{DEL}$ (Blow lab) after 2 min of sperm DNA addition. The extract was optionally supplemented with DMSO and Cullin ligase inhibitor (MLN4924 at 10 $\mu$M).

### Quantification of Western blots

Western blot films from three independent experiments were scanned to generate high-resolution, 300 dpi, 8-bit JPEG images. The pixel intensity of protein bands was then quantified with Image J (http://rsbweb.nih.gov/ij/) and the average intensity (a.u.) calculated for each time point. For quantification of Mcm7 ubiquitylation, a frame was first drawn around the entire ubiquitylation signal to include that which was built in S-phase and that which was extended further in mitosis. This generates a plot for each sample. To then measure only the intensity of ubiquitylation which occurs in mitosis, a line was drawn through the plots to separate the lower region (S-phase) and the upper region (mitosis). The intensity of

this upper region was then measured and a.u. calculated from three independent experiments.

## Immunoprecipitation of post-termination CMG associated with mitotic chromatin

3.75 ml of interphase *X. laevis* egg extract was supplemented with 10 ng/μl of demembranated sperm nuclei, 5 mM caffeine, and 50 μM p97 inhibitor NMS873. The reaction was incubated at 23°C for 90 min to allow completion of DNA replication, after which mitosis was driven by addition of recombinant cyclin A1 NΔ56 at 826 nM followed by incubation at 23°C for a further 60 min. At this stage, chromatin was isolated as described above and chromatin-bound protein complexes released into solution by chromosomal DNA digestion with 2 U/μl benzonase for 15 min. Solubilisation of chromatin-bound protein complexes was further facilitated by subjecting the sample to 5 min of 30 s ON/OFF sonication cycles using a diagenode bioruptor and increasing the concentration of potassium acetate up to 150 mM. The resulting protein complexes were subsequently subjected to either nonspecific IgG (from sheep serum) or Mcm3 immunoprecipitation and the immunoprecipitated material analysed by mass spectrometry as previously described (Sonneville et al, 2017) in collaboration with Dr Richard Jones from MS Bioworks LLC.

## Supplementary Information

## Acknowledgements

Dr S Priego Moreno was funded by Wellcome Trust Institutional Strategic Support Fund (ISSF) Award, Dr RM Jones and Dr A Gambus were funded by MRC CDA MR/K007106/1 and Shaun Scaramuzza by Midlands Integrative Biosciences Training Partnership (MIBTP) studentship. We would like to thank Prof Ron Hay for pET28a-pHISTEV30a-SENP1(415-649) and Prof Julian Blow for pET23a-*X.l.*Cyclin A1 NΔ56 and geminin^DEL.

### Author Contributions

S Priego Moreno: conceptualization, data curation, investigation, and writing—review and editing.
RM Jones: data curation, investigation, methodology, and writing—review and editing.
D Poovathumkadavil: data curation, investigation, and writing—review and editing.
S Scaramuzza: data curation and writing—review and editing.
A Gambus: conceptualization, formal analysis, supervision, funding acquisition, project administration, and writing—original draft, review, and editing.

### Conflict of Interest Statement

The authors declare that they have no conflict of interest.

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
