## [Reviewer comments · Life Science Alliance]

Life Science Alliance

Mitotic replisome disassembly depends on TRAIP ubiquitin ligase activity

Agnieszka Gambus, Rebecca Jones, Shaun Scaramuzza, Divyasree Poovathumkadavil, and Sara Priego Moreno

DOI: <https://doi.org/10.26508/lsa.201900390>

Corresponding author(s): *Agnieszka Gambus, University of Birmingham*

Review Timeline:	Submission Date:	2019-03-26
	Editorial Decision:	2019-03-27
	Revision Received:	2019-03-28
	Accepted:	2019-03-29

Scientific Editor: Andrea Leibfried

Transaction Report:

Please note that the manuscript was previously reviewed at another journal and the reports were taken into account in inviting a revision for publication at *Life Science Alliance* prior to submission to *Life Science Alliance*.

Referee #1 Review

Report for Author:

In this manuscript, the authors used *X. laevis* egg extracts to analyze replisome disassembly in mitosis, and check whether the results are similar to those they previously obtained using *C. elegans* embryos. *X. laevis* egg extracts are a useful system to analyze the processes regulating S phase, and events mimicking mitosis can also be induced. However, the kinetics and the efficiency of the reactions should be tightly controlled to avoid misinterpretations, particularly when the experiments require the addition of several reagents that might dilute the extract. In the analyses described in this manuscript, a number of reagents were added at different steps, and without the required controls, it is difficult to reach firm conclusions. I also found that the main results claimed in this manuscript lack sufficient novelty for publication in a high impact journal.

Other main comments:

It is not clear to me why the authors are using cyclin A Δ instead of (non-degradable) cyclin B. Cyclin A1 Δ is a meiotic cyclin and not a mitotic cyclin, and therefore the ensuing cascade of events might be different than those acting in real mitotic cycles. Consequently, it is difficult to correctly interpret most of the results. The conclusion that mitosis is correctly reproduced, based on EV Fig 1B showing only the condensation of chromosomes, might be erroneous. Moreover, the corresponding panel on chromosome condensation is not very convincing. It is difficult to correctly interpret Figure 1 because there is no control of the unloading reaction. No kinetic is shown with only sperm nuclei in egg extracts to show the fate of CDC 45 and MCMs. In addition, a very low level of MCM2-7 is effectively unloaded in the presence of cyclin A1 Δ . No control of the DNA synthesis kinetics is shown in these experiments. Similarly, controls of the extent of DNA synthesis lack also for Figure 2 that analyses immunoprecipitated MCM3 from chromatin in different samples. The same remark applies for the other figures as well.

Referee #2 Review

Report for Author:

The manuscript by Moreno et al provides a detailed mechanism about how the rest of the replisome that continues to stay on chromatin disassembles in mitosis in their model system of *Xenopus* egg extracts. The authors (lab) previous work (Moreno et al, Science, 2014) defined a novel mechanism involving K48 linkage-dependent ubiquitylation of the MCM helicase subunit, Mcm7, in the termination of DNA replication forks at the end of S-phase. Here, they continue with a similar type of analysis to show that ubiquitylation of Mcm7 involving different linkage-dependency (K6 and K63) drives the dissociation of the replisome (removal of Cdc45) from the mitotic chromatin. In addition, they provide some evidence that the TRAIPIP ubiquitin ligase is responsible for the K6/K63 ubiquitin linkages on Mcm7 that displaces the replisome via the p97 segregase using chemical inhibitors. While the study adds an important step to understanding how polyubiquitylation drives replisome disassembly from chromatin after DNA replication in *Xenopus* egg extracts, the scope of the study remains very limited and does not add whether this mechanism is conserved in mammalian cells.

Major points:

1) While this study nicely characterizes how Cdc45 is removed from mitotic chromatin, the study is largely descriptive, the mechanism that the authors proposed based on their studies could be entirely indirect. There are no interrogation of this proposed mechanism using ubiquitylation-defective (lysine site-specific) mutations on Mcm7 that separates its replicative helicase function from its ability to cooperate with the p97 segregase after replication termination or in mitosis. Also, there are no antibody depletion and add-back experiments done for TRAIP ubiquitin ligase and mutants that prevent its binding or recognition of Mcm7 to definitively show that this ligase works exclusively during mitosis but not at the end of S-phase.

2) Although it is notable that understanding the underlining mechanism of replisome dissociation from the mitotic chromatin could provide valuable insight into how things work in different biological settings; for a broader scope, the authors should at least attempt to figure out whether the same mechanism applies to mammalian cells (or not). For example, does the depletion of TRAIP ubiquitin ligase in human cells affect MCM or Cdc45 loading or chromatin accumulation in different cell cycle phases (under the same conditions using inhibitors)? Can Mcm7 ubiquitylation by K6 or K63 linkages be observed in human cells? Also, what is missing from all of this is the biological consequence of what happens to cells when the replisome can't be efficiently cleared during mitosis or after the end of S-phase. What is the cellular consequence of this? The best way to address this question is to use intact cells.

Referee #3 Review

Report for Author:

This paper looks at how replisomes are disassembled during mitosis, using frog egg extracts as a model system. The backstory is that, in 2014, the Gambus and Labib labs published Science papers showing that polyubiquitylation of CMG triggers replisome disassembly by the p97 AAA+ ATPase during replication termination in late S-phase. Follow up studies by the Labib/Gambus, and Walter labs in 2017 identified CUL-2/LRR1 as the ubiquitin ligase in charge of replisome disassembly. The former study also showed that, in *C. elegans*, a backup pathway operating in mitosis clears replisomes from chromatin in a Cul2-independent manner. The goal of the current study was to determine if this backup, mitotic pathway is present in vertebrates (namely *Xenopus*) and, if so, to understand how it works. The data presented are generally of good quality and the authors do a good job of showing that *Xenopus* does indeed contain a CUL-2 independent pathway at mitosis that disassembles replisomes. The authors go on to search for mediators of this mitotic pathway and they land on the Ub E3 ligase TRAIP. As detailed below, there is one major issue that renders the paper unpublishable in its current form, and there are also a few minor issues the authors may want to take care of.

Major issue:

The authors identify the Ub E3 ligase TRAIP as responsible for MCM7 ubiquitylation in mitosis and, presumably, subsequent replisome disassembly. Surprisingly, however, there are no experiments that directly test the crucial point that loss of TRAIP would prevent replisome

disassembly in mitosis! Regarding TRAIP, a single loss-of-function experiment is presented (Fig 3C), where a purportedly dominant-negative form of the enzyme is added to extract and we see that mitotic ubiquitylation of MCM7 is reduced at the 45' timepoint. This is an odd experiment and it undermines the entire paper. First, a truncated time-course is presented, just 15' and 45', whereas most of the other experiments go out to 75' or 120'. Second, when replisome disassembly is examined, there is no difference between buffer vs WT vs dom-neg TRAIP. So while "inactivation" of TRAIP has an effect on MCM7 ubiquitylation at an early timepoint, it does not impact the replisome. The best way to directly test for a role for TRAIP in mitotic disassembly is to do the standard immuno-depletion and add-back, the bread and butter of the *Xenopus* system, and then the authors will know. This experiment is mandatory for this to be a publishable paper.

Minor issues:

There is no mention of panel D in the EV Fig 1 legend.

In EV Fig 2, SUMO1 sample set, there is a large amount of signal in the "no DNA" lane. Was the blot mislabeled?

The labeling of the figures does not use a consistent nomenclature for the inhibitors. For example, the p97 inhibitor is sometimes called p97i and sometimes NMS873.

Referee #1:

We are glad to see that Referee #1 agrees that “*X. laevis* egg extracts are a useful system to analyse the processes regulating S-phase, and events mimicking mitosis can also be induced”.

Response to further referee comments below:

1. The kinetics and the efficiency of the reactions should be tightly controlled to avoid misinterpretations, particularly when the experiments require the addition of several reagents that might dilute the extract. In the analyses described in this manuscript, a number of reagents were added at different steps, and without the required controls, it is difficult to reach firm conclusions.

We apologise for not including DNA synthesis kinetics for every experiment. The effects of all the inhibitors that we use on DNA synthesis in egg extract have been previously published by us and others. We always induce mitosis by addition of Cyclin A1 Δ after completion of DNA synthesis (there is no more DNA synthesis detectable in the bulk DNA synthesis assay) and so the kinetics of DNA synthesis in mitosis are a flat line.

To address the referee’s comment however we now:

- Show the kinetics of DNA synthesis for extract upon treatment with Cullin inhibitor MLN4924 and optional addition of Cyclin A1 Δ (EV Fig 1A) over a number of repeats showing reproducibility of this observation.
- Show the kinetics of DNA synthesis in S-phase upon treatment with NMS873 with optional addition of Cyclin A1 Δ (EV Fig 2D).
- Show the kinetics of DNA synthesis in S-phase upon treatment with aphidicolin/caffeine with optional addition of Cyclin A1 Δ (EV Fig 6).
- Show that addition of MLN4924 Cullin inhibitor in S-phase does not affect entry into mitosis, using western blotting for mitotic markers phospho Ser10-histone H3 and Smc2 (EV Fig 1C).
- Show that addition of NMS873 segregase inhibitor together with Cyclin A1 Δ does not affect entry into mitosis, using western blotting for the mitotic marker Smc2 (Fig 3B) and microscopy of nuclear membrane degradation (EV Fig 2B).
- Show that addition of UbiNOK polyubiquitylation inhibitor together with Cyclin A1 Δ does not affect entry into mitosis, using microscopy of nuclear membrane degradation (EV Fig 2E).
- Show that addition of recombinant TRAIPwt or TRAIPmut together with Cyclin A1 Δ and/or the p97 inhibitor NMS873 does not influence entry into mitosis using western blotting for the mitotic marker Smc2 (Fig 3C and D) and microscopy of nuclear membrane degradation (EV Fig 3D).

2. I also found that the main results claimed in this manuscript lack sufficient novelty for publication in this journal.

In this manuscript we provide evidence that any replisomes that fail to be unloaded from chromatin in S-phase can be unloaded during mitosis. This occurs through K6/K63-linked ubiquitylation of subunit Mcm7 of the replicative helicase, which is

dependent on the activity of TRAIPI ubiquitin ligase. The novel points of this manuscript are:

- We show that the process of unloading of post-termination replisome in mitosis is conserved in vertebrates, following our initial discovery of this pathway in *C. elegans* embryos.
- We show for the first time that stalled replisomes are also unloaded in mitosis.
- We have identified the ubiquitin ligase TRAIPI as being required for mitotic replisome unloading (both post-termination and stalled).
- We provide the first mechanistic elements of this process – e.g. the type of ubiquitin chains that are built on the replisome in mitosis.
- We describe for the first time the composition of the replisome, which is retained on chromatin through into mitosis, should S-phase disassembly be inhibited.
- We clarify that SUMOylation is not needed for mitotic replisome disassembly in vertebrates.

We believe that the presented work provides an important step in our understanding of replisome transactions and the pathways through which it is processed during different stages of the cell cycle. Such elements make it worthy for publication in this journal.

3. It is not clear to me why the authors are using cyclin A Δ instead of (non-degradable) cyclin B. Cyclin A1 Δ is a meiotic cyclin and not a mitotic cyclin, and therefore the ensuing cascade of events might be different than those acting in real mitotic cycles. Consequently, it is difficult to correctly interpret most of the results.

We agree with the referee that indeed in mammalian cells Cyclin A is a meiotic cyclin, however this differs in our model system *Xenopus laevis* egg extract. Classic work on cyclin/CDKs from Tim Hunt's lab showed that during *Xenopus* oogenesis Cyclin B2 and B1 are expressed at high levels, while Cyclin A1 is expressed at a low level (Kobayashi et al JCB 1991), and it is B type cyclins that drive *Xenopus* oocyte maturation i.e. meiotic divisions (Hochegger et al Development 2001). Cyclin A1 is an embryonic form of Cyclin A, which is synthesised from maternally deposited mRNA and which diminishes around the mid-blastula transition, when zygotic transcription and synthesis of Cyclin A2 begins. The level of maternal mRNA of Cyclin A1 in mature oocytes is similar to the level of B1 and B2 mRNA (Minshull et al EMBO J 1990) and *Xenopus* egg extract made from eggs actively synthesises Cyclin A1, B1 and B2 (Minshull et al Cell 1989). Importantly, it has been shown that both Cyclin A1 and Cyclin B1 show similar histone H1 kinase activity in egg extract and that both, when added to egg extract as recombinant proteins, can stimulate progression of extract into mitosis at the same concentration (Strausfeld et al JCS 1996). The regulation of cell cycle progression differs therefore between the embryonic setup of egg extract and that of somatic cells and is much more flexible in the egg extract. For this reason, many scientists studying mitotic transactions in egg extract use the extract prepared from eggs arrested in metaphase of meiosis II (e.g. Shintomi et al Science 2017).

During embryogenesis, Cyclin A1 is degraded earlier in mitosis than Cyclin B1 and B2 (Minshull et al EMBO J 1990). We have decided to use Cyclin A1 N Δ 56 as this construct has been used previously to successfully induce mitosis in egg extract (Strausfeld et al JCS 1996) and, as this mutant of Cyclin A1 cannot be degraded, the extract stays arrested in mitosis rather than moving to the next stage of the cell cycle, which would desynchronise the system.

In response to this comment, we now include within the manuscript a more thorough explanation of Cyclin/CDK regulation in *Xenopus* egg extract and the choice of Cyclin in our experiments (page 3 of the manuscript). We also provide, in a number of ways, evidence for induction of mitosis in our system – please see below.

4. The conclusion that mitosis is correctly reproduced, based on EV Fig 1B showing only the condensation of chromosomes, might be erroneous. Moreover, the corresponding panel on chromosome condensation is not very convincing.

We have now included a more thorough explanation of the experimental setup (page 3 and 4) and provide evidence for mitosis induction in our system in the following ways:

- We show by DIC and Dil staining of nuclear membranes that the nuclear envelope efficiently breaks down upon Cyclin A1 Δ addition (EV Fig 1B).
- We observe chromosome condensation more clearly by microscopy (EV Fig 1B).
- We show phosphorylation of Serine 10 on histone H3 upon addition of Cyclin A1 Δ , which coincides with chromosomes condensation (EV Fig 1C).
- We show recruitment of condensin Smc2 to chromatin upon Cyclin A1 Δ addition (EV Fig 1C).

5. It is difficult to correctly interpret Figure 1 because there is no control of the unloading reaction. No kinetic is shown with only sperm nuclei in egg extracts to show the fate of CDC 45 and MCMs.

We have now provided the chromatin isolation control for this experiment without cullin inhibitor i.e. only sperm nuclei addition (Fig 1B). In the majority of our reactions, we isolate chromatin at times when replication is long finished in control conditions and so there are no replication factors binding to chromatin and the western blots are blank. This is the reason we did not include them in the original version, but we happily include them now. We also now provide a measurement of DNA synthesis kinetics in control conditions (EV Fig 1A).

6. a very low level of MCM2-7 is effectively unloaded in the presence of cyclin A1 Δ .

We now include an explanation in the manuscript as to why this is the case (page 4/5) – because of the high quantity of DNA used in our experiments we have a proportion of nuclei that do not replicate, but they do have licensed origins and as such we can detect large quantities of dormant, inactive Mcm2-7 double-hexamers on chromatin, which we have shown previously are not unloaded through the Mcm7 polyubiquitylation pathway and require active replication to disassemble (Moreno et al Science 2014). We now also provide an experiment with a reduced level of DNA and minimal licensing of origins to show that under conditions of restricted Mcm2-7 loading, when most of the Mcm2-7 complexes become activated and turned into CMGs, we do see a reduction in the signal of not only Cdc45 and Psf2 but also Mcm7 during mitosis (EV Fig 2A).

7. No control of the DNA synthesis kinetics is shown in these experiments.

Most of our experiments are carried out after replication is complete and so there is no more DNA synthesis to detect when we induce mitosis. We now provide a replication synthesis kinetics experiment for extract upon treatment with the Cullin inhibitor

MLN4924, with optional addition of Cyclin A1 Δ (EV Fig 1A), showing reproducibility of this observation. We also show the DNA synthesis kinetics for S-phase upon treatments with NMS873 and aphidicolin/caffeine with optional addition of Cyclin A1 Δ (EV Fig 2D and EV Fig 6).

8. Similarly, controls of the extent of DNA synthesis lack also for Figure 2 that analyses immunoprecipitated MCM3 from chromatin in different samples. The same remark applies for the other figures as well.

We have now added a DNA synthesis kinetics control for the experiment in Figure 2 (EV Fig 3A). As explained above we have now also provided a DNA synthesis kinetics control experiment for all of the treatments added during S-phase, which may change the ability of the extract to synthesise DNA. All of the inhibitors that we add at the same time as Cyclin A1 Δ are added after completion of DNA synthesis. For each of these treatments we have therefore instead provided controls to show that they do not affect the ability of the extract to enter mitosis i.e. western blotting of the mitotic marker Smc2 and/or analysis of nuclear membrane breakdown through microscopy (Fig 3B, 3C, 3D, EV Fig 1C, 2B, 2E, 3D).

Referee #2:

We would like to thank Referee #2 for stating that: “the study adds an important step to understanding how polyubiquitylation drives replisome disassembly from chromatin after DNA replication in *Xenopus* egg extracts”, and we have carried out the following improvements to address the referee’s comments:

Response to further referee comments below:

- 1) While this study nicely characterizes how Cdc45 is removed from mitotic chromatin, the study is largely descriptive, the mechanism that the authors proposed based on their studies could be entirely indirect. There are no interrogation of this proposed mechanism using ubiquitylation-defective (lysine site-specific) mutations on Mcm7 that separates its replicative helicase function from its ability to cooperate with the p97 segregase after replication termination or in mitosis.

We agree with the referee that the above experiments would be very valuable. The problem with delivery of them is that the mutant of Mcm7 that cannot be ubiquitylated does not exist at present. Potential ubiquitylation sites within Mcm7 have been identified in *S. cerevisiae in vitro*, but mutating them does not stop Mcm7 ubiquitylation *in vivo* (Maric et al., Cell Rep 2017). We have also mapped likely ubiquitylation sites within *Xenopus* Mcm7 but again their mutation does not prevent Mcm7 ubiquitylation. Ubiquitylation as a modification is known to jump to a different lysine when the preferred one is not available, while mutating multiple lysines renders the Mcm7 protein inactive. Screening different mutations in Mcm7 in *Xenopus* extract is also not straightforward as Mcm2-7 complexes containing each mutant Mcm7 need to be expressed and purified from insect cells and added to Mcm2-7 depleted extract. This type of work is therefore easier to carry out in different model systems.

Having said this however, the *Xenopus* system allows us to carry out much more specific inhibition of particular enzymes. There are no other organelles within this system, apart from nuclei, and ATP for the reaction is provided externally. When sperm DNA is added to the extract, it decondenses before the nuclear envelope forms around it and a whole round of replication takes place in the absence of transcription. DNA replication is therefore the main activity of the extract. This reductionist system argues therefore that the effects we observe on DNA replication upon inhibition of particular enzymes or processes are most likely direct and we now provide many more controls to show that our treatments do not inhibit either progression of S-phase or entry into mitosis.

- 2) Also, there are no antibody depletion and add-back experiments done for TRAIIP ubiquitin ligase and mutants that prevent its binding or recognition of Mcm7 to definitively show that this ligase works exclusively during mitosis but not at the end of S-phase.

'no antibody depletion and add-back experiments':

We agree that immunodepletion experiments to analyse TRAIIP in this process are valuable; please see our in depth response below for comment 2, Referee #3.

'ligase works exclusively during mitosis':

We have now added a control chromatin isolation experiment to show that when Cullin activity is blocked we do not see high levels of Mcm7 modification in late S-phase, nor much unloading of CMG helicase (Fig 1B). It is likely that TRAIIP is active in S-phase, for example in helping replisomes to pass barriers such as inter-strand crosslinks (ICLs), but TRAIIP activity is not essential for Mcm7 polyubiquitylation and unloading in unperturbed S-phase (see Rev Fig 1). We do not state that TRAIIP works exclusively in mitosis, rather that its activity is required for mitotic replisome unloading.

- 3) Although it is notable that understanding the underlining mechanism of replisome dissociation from the mitotic chromatin could provide valuable insight into how things work in different biological settings; for a broader scope, the authors should at least attempt to figure out whether the same mechanism applies to mammalian cells (or not). For example, does the depletion of TRAIIP ubiquitin ligase in human cells affect MCM or Cdc45 loading or chromatin accumulation in different cell cycle phases (under the same conditions using inhibitors)? Can Mcm7 ubiquitylation by K6 or K63 linkages be observed in human cells?

We completely agree with the referee that it would be very valuable to understand whether this mitotic replisome disassembly works in human cells. We are attempting to look at this but currently it is beyond the scope of this manuscript.

This manuscript focuses on the mitotic back-up pathway of replisome disassembly; it is very difficult to address this in human cells because the main S-phase pathway of replisome disassembly is not as yet characterised in human cells and there are no established tools. We cannot simply use the inhibitors we use in *Xenopus* extract due to multiple effects they exert in human cells. For example, using MLN4924 in human cells leads to Cdt1 stabilisation, re-replication and arrest of cells in S-phase, meaning we cannot then study the cells in mitosis. The same is not a problem in the *Xenopus* system because to induce re-replication in this embryonic system we need to not only block

degradation of Cdt1 but also supplement the extract with recombinant Cdt1. Similarly, p97 segregase has a whole myriad of functions in human cells and inhibition of p97 therefore leads to disruption of many cellular processes and stimulates DNA damage. As much as we agree that elucidating these pathways in human cells is important, we would also like to point out that all of the work done and published so far about replication termination has been carried out in model systems (Maric et al Science 2014; Moreno et al Science 2014; Dewar et al Nature 2015; Sonnevile et al Nature Cell Biology 2017; Dewar et al Genes & Dev 2017), while Prof Walter's group has at present a manuscript under revision in Cell describing replisome unloading in mitosis, which is delivered entirely in the *Xenopus* system (Deng et al bioRxiv 2018).

- 4) Also, what is missing from all of this is the biological consequence of what happens to cells when the replisome can't be efficiently cleared during mitosis or after the end of S-phase. What is the cellular consequence of this? The best way to address this question is to use intact cells.

We agree with the referee completely – it is important to address the consequences of disrupting these processes and that the best way to do this is with intact cells. Indeed once we have characterised such processes in human cells, we plan to address this. Due to the points raised above however, we believe that such work is beyond the scope of this current manuscript.

Referee #3:

We are very happy to see that Referee #3 believes that “the data presented are generally of good quality and the authors do a good job of showing that *Xenopus* does indeed contain a CUL-2 independent pathway at mitosis that disassembles replisomes”. Thank you.

Response to further referee comments below:

1. The authors identify the UB E3 ligase TRAIP as responsible for MCM7 ubiquitylation in mitosis and, presumably, subsequent replisome disassembly. Surprisingly, however, there are no experiments that directly test the crucial point that loss of TRAIP would prevent replisome disassembly in mitosis! Regarding TRAIP, a single loss-of-function experiment is presented (Fig 3C), where a purportedly dominant-negative form of the enzyme is added to extract and we see that mitotic ubiquitylation of MCM7 is reduced at the 45' timepoint. This is an odd experiment and it undermines the entire paper. First, a truncated time-course is presented, just 15' and 45', whereas most of the other experiments go out to 75' or 120'.

We now present further data showing that TRAIP activity is needed for unloading of replisomes in mitosis (Fig 3C and 5C) and for mitotic ubiquitylation of MCM7 in both post-termination (Fig 3D) and stalled replisomes (Fig 5D). These experiments were performed by adding a high concentration of wt and catalytically dead (C25A) TRAIP to the egg extract to compete with endogenous TRAIP. All of these experiments now also

have a consistent set of time points. Moreover, we have quantified our results over three independent repeats of experiments in Figure 3 to indicate their high reproducibility. We also present an *in vitro* autoubiquitylation assay of TRAIP to show that the C25A mutant is indeed ubiquitin ligase dead (EV Fig 3C).

2. Second, when replisome disassembly is examined, there is no difference between buffer vs WT vs dom-neg TRAIP. So while "inactivation" of TRAIP has an effect on MCM7 ubiquitylation at an early timepoint, it does not impact the replisome. The best way to directly test for a role for TRAIP in mitotic disassembly is to do the standard immuno-depletion and add-back, the bread and butter of the *Xenopus* system, and then the authors will know. This experiment is mandatory for this to be a publishable paper.

The experiment we presented in our previous version of this manuscript was indeed interrogating only the ubiquitylation of Mcm7 and not replisome disassembly. That was the case because we added the inhibitor of p97 segregase NMS873 to the reaction in mitosis. Such a treatment allowed us to visualise ubiquitylated Mcm7 still on chromatin. We apologise if it was not explained clearly enough.

Now we present a set of two experiments for each: post-termination and stalled replisomes. One, without p97 inhibitor, to assess replisome unloading and the second, with p97 inhibitor, to assess Mcm7 ubiquitylation on chromatin. Both were carried out with addition of a high concentration of the wt or ligase dead C25A mutant of TRAIP to compete with endogenous TRAIP.

We agree that immunodepletion experiments are the traditional way of showing the requirement of a particular protein for a particular process in the *Xenopus* egg extract system. However, the real strength in them comes from being able to rescue the immunodepletion with recombinant wt protein – and the best result from not being able to rescue them with an inactive mutant. Without being able to rescue the phenotype by “adding back”, immunodepletion in fact comes with high risks of artefacts. What we provide in this manuscript is the “add in” version of this, without the immunodepletion.

To achieve immunodepletion in the extract we need very good antibodies that can very efficiently and specifically immunoprecipitate TRAIP. We have tested a number of antibodies against TRAIP for this ability and then went on to test those which worked for immunodepletion. The first (NBP1-87125) produced no detectable immunodepletion at all (see Rev Fig 2), whilst the second (Abxexa 238924) produced about 40% depletion (Rev Fig 3 (top left)), using our standard immunodepletion procedure. Unfortunately this was not enough to reduce TRAIP chromatin binding in mitosis (Rev Fig 3 (top right)). To improve this, we increased the number of rounds of depletion using the same Abxexa 238924 antibody and in doing so we did achieve a much better depletion (Rev Fig 3 (bottom left)), but unfortunately this led to a reduced ability of the extract to replicate DNA in interphase (Rev Fig 3 (bottom middle)) and therefore very little replisomes left to observe in mitosis (Rev Fig 3 (bottom right)). It is possible that immunodepletion of TRAIP from interphase extract indeed inhibits DNA replication, which would render studying the mitotic role of TRAIP very difficult indeed: we would need to replicate DNA in normal extract and then transfer the DNA to the immunodepleted extract, already induced to go into mitosis. In this case we may then have the problem that TRAIP from the first extract remains associated with post-termination replisomes from late S-phase and so could drive their disassembly in

mitosis. Alternatively, the 3 rounds of depletion with our antibody created too much stress on the extract, rendering it incapable of DNA replication. To get meaningful results with immunodepletion we therefore need better antibodies, which we do not have at present. In having such antibodies, we would still then need to rescue the effects by adding back the same set of recombinant wt and mutant TRAIP we use now to the immunodepleted extract, to show that they work in the same way as in our “add in” experiments. We believe therefore, that add-in experiments are as valuable as immunodepletion/add-back because we can clearly see that the C25A mutant of TRAIP acts as a dominant negative mutant when added to the extract.

3. There is no mention of panel D in the EV Fig 1 legend.

We apologise for the omission, it is now fixed.

4. In EV Fig 2, SUMO1 sample set, there is a large amount of signal in the "no DNA" lane. Was the blot mislabeled?

Thank you for pointing this out. We now provide a blot of a different repeat of this experiment without the unusual lane – now (EV Fig 4).

5. The labeling of the figures does not use a consistent nomenclature for the inhibitors. For example, the p97 inhibitor is sometimes called p97i and sometimes NMS873.

We have unified our labelling. Thank you for pointing this out to us.

Rev Fig 1. TRAIPlwt or TRAIPlmut were optionally added to egg extract at the same time as sperm DNA and replication reaction run in their presence. Chromatin was isolated at indicated timepoints and Mcm7 ubiquitylation and replisome disassembly analysed by western blotting. Mcm7 is ubiquitylated and replisome disassembled (Cdc45) equally efficiently in the presence of TRAIPlmut as in control.

Rev Fig 2. Immunodepletion of TRAIPl with NBP118725 antibody. There is no noticeable immunodepletion.

Rev Fig 3. Immunodepletion of TRAIPl with Abxexa 238924 antibody. With two rounds of depletion about 40% of TRAIPl was depleted but that was not enough as still normal level of TRAIPl binds to chromatin in mitosis. With three rounds of depletion we depleted most of the protein, which affected efficiency of DNA replication in interphase and massively reduced the level of retained replisomes until mitosis.

Referee #1 Review

Report for Author:

The authors have relatively answered the questions I suggested in my report, and clarified some points. However, it is not clear to me if the kinetics of DNA synthesis shown as a supplementary material strictly correspond to the experiments they showed in the first version of the paper. As I already mentioned, in the analyses described in this manuscript, a number of reagents were added at different steps, and without the required parallel controls, it is difficult to reach firm conclusions.

Moreover, I am only partly convinced by the answers given to the other reviewers, and therefore I think that it is difficult to support the publication of this manuscript.

Referee #2 Review

Report for Author:

Manuscript by Moreno et al is a revised manuscript that has previously been reviewed. Although the authors have addressed some of the technical points raised by the reviewers, the major criticism of the manuscript brought up by Reviewers #2 and #3 was not addressed, which is to show that the TRAIP ubiquitin ligase is genetically important/relevant for mitotic replisome disassembly in vertebrate cells (mammalian cells). Without add-back reconstitution experiments in *Xenopus laevis* egg extract, the use of dominant negative TRAIP (add-in) is insufficient to justify their conclusion that TRAIP is the ubiquitin ligase that polyubiquitinates MCM7 to trigger mitotic disassembly. Using overexpressed catalytically-dead ubiquitin E3 ligase proteins could non-specifically affect binding and sequester other factors that may be critical for replisome disassembly and the ubiquitin conjugation pathway.

Minor points:

1) The manuscript is written like a review article, the title "Mitotic replisome disassembly in vertebrates" is quite broad for what they show using *Xenopus laevis* egg extracts. Also to use "Recent years have brought a breakthrough..." in their abstract is a bit inappropriate since the study will be dated very soon. The authors also keep mentioning that the mitotic replisome disassembly pathway is conserved throughout evolution in higher eukaryotes (vertebrates), I'm not sure what this means. If this is not present in *S. cerevisiae*, but is present in *Xenopus laevis* egg extract, that doesn't mean it is conserved in higher eukaryotes. If they see the same requirement of mitotic disassembly in *C. elegans*, then they should knockout/inactivate Traip in *C. elegans* and do some functional studies there. For example, they didn't show that TRAIP is working in mammalian cells or chicken cells in the same manner as in frog extracts. etc... Basically, the experiments were done well, but the manuscript is filled with over-interpretation of the significance of their data.

2. The authors should show that K6 and K63 sites are critical and necessary for Mcm7 polyubiquitination. They need to use a ubiquitin that is mutated on both of those sites (other sites should be WT), and show that this cannot support any Mcm7 polyubiquitination and disassembly in their in vitro assay.

Referee #3 Review

Report for Author:

The major critique from my initial review was that the authors lacked convincing evidence that TRAIP is indeed required for replisome removal during mitosis. My suggestion was a simple immunodepletion & add-back experiment. Unfortunately, no such experiment is included in the revised manuscript. In their rebuttal, the authors state that their antibodies are not good enough to allow such an experiment. Instead, the authors add high concentrations of a dominant-negative form of TRAIP and observe replisome retention on chromatin in a mitotic extract. As over-expression experiments are prone to artifacts this is, in my view, not the cleanest way to make this important point. Another relevant factor in all of this is a recently published Molecular Cell paper by the Pellman and Walter labs that uses *Xenopus* egg extracts to examine replisome disassembly during mitosis. In this paper the authors were able to efficiently deplete TRAIP from extracts and they observed replisome retention during mitosis. In addition, they could reverse this effect by add-back of the WT but not a mutant form of the protein. These data show that the requested experiment is indeed technically feasible.

March 27, 2019

RE: Life Science Alliance Manuscript #LSA-2019-00390-T

Dr. Agnieszka Gambus
University of Birmingham
School of Cancer Sciences
Chromosomal Replication Laboratory
Edgbaston
Birmingham B15 2TT
United Kingdom

Dear Dr. Gambus,

Thank you for transferring your revised manuscript entitled "Mitotic replisome disassembly in vertebrates" to Life Science Alliance. Your manuscript was reviewed twice at another journal before, and the editors transferred those reports to us with your permission.

The reviewers thought that firm evidence for TRAIP being required for replisome removal during mitosis was not provided and they would have expected an add-back experiment to better support your conclusions. Despite the lack thereof, we would be happy to publish your paper in Life Science Alliance pending final revisions necessary to address the remaining reviewer concerns and to meet our formatting guidelines:

- please follow reviewer #2's suggestion for re-writing
- please acknowledge the lack of a proper add-back experiment in the manuscript text
- while reviewer #2's suggestion for testing the importance of K6 and K63 for poly-ubiquitination is a good one, this request was made only in the second round of review and can thus be only responded to in a point-by-point response
- please note that we only have Supplementary figures at LSA (not EV figures)
- please provide source data for figure 1C and current EV2C
- please note that current EV3E is not called-out in the text

A. FINAL FILES:

B. MANUSCRIPT ORGANIZATION AND FORMATTING:

Thank you for your attention to these final processing requirements.

Sincerely,

Andrea Leibfried, PhD
Executive Editor
Life Science Alliance

Meyerhofstr. 1
69117 Heidelberg, Germany
t +49 6221 8891 502
e a.leibfried@life-science-alliance.org
www.life-science-alliance.org

Response to reviewer comments:

1. Suggestions for rewriting:

The manuscript is written like a review article, the title "Mitotic replisome disassembly in vertebrates" is quite broad for what they show using *Xenopus laevis* egg extracts.

We have now changed the title to be more specific:

"Mitotic replisome disassembly depends on TRAIP ubiquitin ligase activity".

Also to use "Recent years have brought a breakthrough..." in their abstract is a bit inappropriate since the study will be dated very soon.

The mechanism of replication termination was first reported in 2014, but we have now eliminated this opening sentence.

The authors also keep mentioning that the mitotic replisome disassembly pathway is conserved throughout evolution in higher eukaryotes (vertebrates), I'm not sure what this means. If this is not present in *S. cerevisiae*, but is present in *Xenopus laevis* egg extract, that doesn't mean it is conserved in higher eukaryotes. If they see the same requirement of mitotic disassembly in *C. elegans*, then they should knockout/inactivate Traip in *C. elegans* and do some functional studies there. For example, they didn't show that TRAIP is working in mammalian cells or chicken cells in the same manner as in frog extracts. etc... Basically, the experiments were done well, but the manuscript is filled with over-interpretation of the significance of their data.

We have eliminated: "The mitotic replisome disassembly pathway is therefore conserved throughout evolution in higher eukaryotes" from the abstract.

We have now exchanged both mentions of vertebrates in the text (on page 2 and page 10) with *Xenopus laevis* egg extract.

We have eliminated words "conserved throughout evolution" on page 2.

We have checked throughout the discussion for over-interpretation of the significance of our data but believe we are rather careful with the language. A few examples are below.

"It is likely that these unreplicated DNA fragments must be processed in mitosis to ensure correct chromosome segregation and this processing will involve replisome unloading and fork remodelling – hence the need for a process of replisome disassembly in mitosis."

"it is interesting to speculate that TRAIP can play an analogous role during DNA damage repair"

"Our data is consistent with a model in which TRAIP drives mitotic replisome disassembly by promoting Mcm7 modification with K6- and K63-linked ubiquitin chains."

2. please acknowledge the lack of a proper add-back experiment in the manuscript text

We have now added the below sentence on page 8:

As we were unable to immunodeplete TRAIP from the egg extract with any of the antibodies tested, we decided to use a dominant-negative, ligase-dead mutant of TRAIP to out-compete the endogenous TRAIP.

3. while reviewer #2's suggestion for testing the importance of K6 and K63 for poly-ubiquitination is a good one, this request was made only in the second round of review and can thus be only responded to in a point-by-point response

We agree that this is an interesting experiment but it was not requested in the previous revision.

4. please note that we only have Supplementary figures at LSA (not EV figures)

We have changed all EV Fig to Supp Fig.

5. please provide source data for figure 1C and current EV2C

These are now provided in a separate document.

6. please note that current EV3E is not called-out in the text

Figure EV3E is called-out in the legend of Figure 3. It explains how we quantified the data in Fig 3D.

March 29, 2019

RE: Life Science Alliance Manuscript #LSA-2019-00390-TR

Dr. Agnieszka Gambus
University of Birmingham
School of Cancer Sciences
Chromosomal Replication Laboratory
Edgbaston
Birmingham B15 2TT
United Kingdom

Dear Dr. Gambus,

Thank you for submitting your Research Article entitled "Mitotic replisome disassembly depends on TRAIIP ubiquitin ligase activity". I appreciate the introduced changes and it is a pleasure to let you know that your manuscript is now accepted for publication in Life Science Alliance. Congratulations on this interesting work.

*****IMPORTANT:** If you will be unreachable at any time, please provide us with the email address of an alternate author. Failure to respond to routine queries may lead to unavoidable delays in publication.*******

DISTRIBUTION OF MATERIALS:

Again, congratulations on a very nice paper. I hope you found the review process to be constructive and are pleased with how the manuscript was handled editorially. We look forward to future exciting

submissions from your lab.

Sincerely,
